# Identifying the biological control of the annual and multi-year variations in South Atlantic air-sea $CO_2$ flux

Daniel J. Ford[1,2], Gavin H. Tilstone[1], Jamie D. Shutler[2] and Vassilis Kitidis[1]

[1] Plymouth Marine Laboratory, Plymouth, UK

[2] College of Life and Environmental Sciences, University of Exeter, Penryn, UK

*Correspondence to*: Daniel Ford (dfo@pml.ac.uk)

**Abstract.** The accumulation of anthropogenic $CO_2$ emissions in the atmosphere has been buffered by the absorption of $CO_2$ by the global ocean which acts as a net $CO_2$ sink. The $CO_2$ flux between the atmosphere and the ocean, that collectively results in the oceanic carbon sink, is spatially and temporally variable, and fully understanding the driving mechanisms

behind this flux is key to assessing how the sink may change in the future. In this study a time series decomposition analysis was applied to satellite observations to determine the drivers that control the sea-air difference of $CO_2$ partial pressure ($\Delta p CO_2$) and the $CO_2$ flux on seasonal and interannual time scales in the South Atlantic Ocean. Linear trends in $\Delta p CO_2$ and the $CO_2$ flux were calculated to identify key areas of change.

Seasonally, changes in both the $\Delta p CO_2$ and $CO_2$ flux were dominated by sea surface temperature (SST) in the subtropics

(north of 40 °S) and were correlated with biological processes in the subpolar regions (south of 40° S). In the Equatorial Atlantic, analysis of the data indicated that biological processes are likely a key driver, as a response to upwelling and riverine inputs. These results highlighted that seasonally $\Delta p CO_2$ can act as an indicator to identify drivers of the $CO_2$ flux. Interannually, the SST and biological contributions to the $CO_2$ flux in the subtropics were correlated with the Multivariate ENSO Index (MEI) which leads to a weaker (stronger) $CO_2$ sink in El Niño (La Niña) years.

The 16-year time-series identified significant trends in $\Delta p CO_2$ and $CO_2$ flux, however, these trends were not always consistent in spatial extent. Therefore, predicting the oceanic response to climate change requires the examination of $CO_2$ flux rather than $\Delta p CO_2$. Positive $CO_2$ flux trends (weakening sink for atmospheric $CO_2$) were identified within the Benguela upwelling system, consistent with increased upwelling and wind speeds. Negative trends in the $CO_2$ flux (intensifying sink for atmospheric $CO_2$) offshore into the South Atlantic Gyre, were consistent with an increase in the export of nutrients from

mesoscale features, which drives the biological drawdown of $CO_2$. These multi-year trends in the $CO_2$ flux indicate that the biological contribution to changes in the air-sea $CO_2$ flux cannot be overlooked when scaling up to estimates of the global ocean carbon sink.

# 1 Introduction

Since the industrial revolution, anthropogenic $CO_2$ emissions have increased unabated and continue to rise atmospheric $CO_2$ concentrations (IPCC, 2021). The global oceans have buffered the rise by acting as a sink for atmospheric $CO_2$ at a rate of between 1 and 3.5 Pg C yr$^{-1}$ (e.g. Friedlingstein et al., 2020; Landschützer et al., 2014; Watson et al., 2020). The strength of the ocean as a sink for $CO_2$ appears to be increasing with time (Friedlingstein et al., 2020; Watson et al., 2020). Regionally this can vary hugely however, and the ocean can oscillate between a source or sink of atmospheric $CO_2$. The difference in the partial pressure of $CO_2$ ($pCO_2$) between the seawater and atmosphere ($\Delta pCO_2$) is used as an indicator or proxy for the net direction of air-sea $CO_2$ flux during gas exchange.

In the open ocean, changes in physical and biogeochemical processes that control seawater $pCO_2$ ($pCO_{2\,(sw)}$) also modify $\Delta pCO_2$ as the atmospheric $pCO_2$ ($pCO_{2\,(atm)}$) is less variable (e.g. Henson et al., 2018; Landschützer et al., 2016). $\Delta pCO_2$ can therefore be controlled by changes in sea surface temperature (SST), because the $pCO_2$ is proportional to the temperature. In addition, plankton net community production (NCP) modifies the concentration of $CO_2$ in the seawater depending on the balance between net primary production (NPP; uptake of $CO_2$ via photosynthesis) and respiration (release of $CO_2$ into the water). The NCP describes the overall metabolic balance of the plankton community, where positive (negative) NCP indicates a drawdown (or release) of $CO_2$ from (or into) the water contributing to a decrease (increase) in $\Delta pCO_2$. Physical processes, including riverine input (e.g. Ibánhez et al., 2016; Lefèvre et al., 2020; Valerio et al., 2021), and upwelling (e.g. González-Dávila et al., 2009; Lefèvre et al., 2008; Santana-Casiano et al., 2009) can alter $pCO_{2\,(sw)}$ and $\Delta pCO_2$ directly through the entrainment of high-$CO_2$ water or indirectly by modifying NCP through nutrient supply (enhancing photosynthesis) and/or organic material supply (enhancing respiration).

The air-sea $CO_2$ flux is more precisely a function of the difference in $CO_2$ concentrations across the mass boundary layer at the ocean's surface, with any turbulent exchange characterised by the gas transfer velocity. The $CO_2$ concentration difference is determined by the $pCO_2$ at the base ($pCO_{2\,(sw)}$) and top ($pCO_{2\,(atm)}$) of the mass boundary layer and the respective solubilities (Weiss, 1974), and must be carefully calculated due to vertical thermo-haline gradients existing across the mass boundary layer (Woolf et al., 2016). The gas transfer velocity is usually parameterised as a function of wind speed (e.g. Ho et al., 2006; Nightingale et al., 2000; Wanninkhof, 2014) which accounts for ~75% of the variance in surface turbulent exchange (e,g, Dong et al., 2021; Ho et al., 2006). Therefore, both oceanographic and meteorological conditions are able to modify and control the seasonality, interannual variability and multi-year trends of this flux.

Seasonal drivers of $\Delta pCO_2$ have been explored globally (Takahashi et al., 2002), and regionally in the Atlantic Ocean (Landschützer et al., 2013; Henson et al., 2018). Takahashi et al. (2002) used binned *in situ* $pCO_{2\,(sw)}$ observations to a 4º by 5º global grid, and found that SST drives $\Delta pCO_2$ in the subtropics, and non-SST processes (i.e. biological activity and ocean circulation) dominate in subpolar and equatorial regions. Landschützer et al. (2013) used a self-organising map feed forward neural network (SOM-FNN) technique to extrapolate the *in situ* $pCO_{2\,(sw)}$ observations and reported similar seasonal drivers in the Atlantic Ocean with one exception, that SST and non-SST processes compensated each other in the Equatorial

Atlantic. Henson et al. (2018) using binned *in situ* observations for the North Atlantic Ocean, also indicated that the subtropics are driven by SST and that subpolar regions are correlated with biological activity.

The interannual drivers of $\Delta p CO_2$ are different compared to the seasonal drivers in the North Atlantic (Henson et al., 2018), which could be true of the South Atlantic Ocean, though this needs to be further investigated. Landschützer et al. (2016,

2014) postulated the El Niño cycle may influence $\Delta p CO_2$ in the subtropical South Atlantic but did not explore the underlying processes. South of 35° S, Landschützer et al. (2015) indicated that atmospheric forcing could control the interannual variability of $\Delta p CO_2$ through changes in Ekman transport and upwelling. These interannual drivers of $\Delta p CO_2$ and the $CO_2$ flux in the South Atlantic Ocean are poorly understood but have key implications for determining how the oceanic $CO_2$ sink could be impacted by climate change and its evolution over interannual and decadal timescales.

In this study, we investigate the drivers of $\Delta p CO_2$ and the $CO_2$ flux in the South Atlantic Ocean over both seasonal and interannual timescales using a timeseries decomposition approach. Trends in $\Delta p CO_2$ and the $CO_2$ flux were calculated from 2002 to 2018, and regions in the South Atlantic Ocean showing the greatest change in the $CO_2$ flux are investigated.

## 2. Data and Methods

### 2.1. $p CO_2$ data

Satellite estimates of $p CO_2$ (sw) were retrieved from the South Atlantic Feed Forward Neural Network (SA-FNN) dataset (Ford et al., 2022, 2021a). Ford et al. (2022) showed that the SA-FNN improved on estimating the seasonal $p CO_2$ (sw) variability in the South Atlantic Ocean compared to the current 'state of the art' methodology (the SOM-FNN). The SA-FNN estimates $p CO_2$ (sw) by clustering *in situ* monthly 1° gridded Surface Ocean $CO_2$ Atlas (SOCAT) v2020 $p CO_2$ (sw) observations (Bakker et al., 2016; Sabine et al., 2013), that have been reanalysed into a dataset configured using consistent

depth and temperature fields (Goddijn-Murphy et al., 2015; Woolf et al., 2016; Reynolds et al., 2002), into eight static provinces in the South Atlantic Ocean (Fig. B1a). The use of eight static provinces allows the SA-FNN to more accurately reproduce the $p CO_2$ (sw) variability. The nonlinear relationships between $p CO_2$ (sw) and three environmental drivers (SST, NCP and $p CO_2$ (atm)), were constructed for each province with a feed forward neural network (FNN). The FNN for each province were applied to produce spatially and temporally complete $p CO_2$ (sw) fields on monthly 1° grids between July 2002 and

December 2018, with uncertainties generated on a per pixel basis as described in Ford et al. (2022). These per pixel uncertainties are shown in Appendix B (Fig. B1).

Monthly 1° grids of $p CO_2$ (atm) were extracted from v5.5 of the global estimates of $p CO_2$ (sw) dataset (Landschützer et al., 2017, 2016) which was calculated using the dry mixing ratio of $CO_2$ from the NOAA-ESRL marine boundary layer reference (https://www.esrl.noaa.gov/gmd/ccgg/mbl/; last accessed 25/09/2020), Optimum Interpolated SST (Reynolds et al., 2002)

and sea level pressure following Dickson et al. (2007). $\Delta p CO_2$ was calculated from $p CO_2$ (sw) and $p CO_2$ (atm) as;

$$\Delta p CO_2 = p CO_2 \text{ (sw)} - p CO_2 \text{ (atm)} \tag{1}$$

## 2.2. Air-sea CO₂ flux data

The air-sea $CO_2$ flux (F) can be estimated using a bulk parameterisation as:

$$F = k \left( \alpha_W \, pCO_{2\,(sw)} - \alpha_s \, pCO_{2\,(atm)} \right) \qquad (2)$$

Where k is the gas transfer velocity which was estimated from ERA5 monthly reanalysis wind speed (Hersbach et al., 2019) following the parameterisation of Nightingale et al. (2000). $\alpha_w$ and $\alpha_s$ are the solubility of $CO_2$ at the base and top of the mass boundary layer at the sea surface (Woolf et al., 2016). $\alpha_w$ was calculated as a function of SST and sea surface salinity (SSS) (Weiss, 1974) using the monthly Optimum Interpolated SST (Reynolds et al., 2002) and SSS from the Copernicus Marine Environment Modelling Service global ocean physics reanalysis product (GLORYS12V1; CMEMS, 2021). $\alpha_s$ was calculated using the same temperature and salinity datasets but included a gradient from the base to the top of mass boundary layer of -0.17 K (Donlon et al., 1999) and +0.1 salinity units (Woolf et al., 2016). $pCO_{2\,(atm)}$ was calculated using the dry mixing ratio of $CO_2$ from the NOAA-ESRL marine boundary layer reference, Optimum Interpolated SST (Reynolds et al., 2002) applying a cool skin bias (0.17K; Donlon et al., 1999) and sea level pressure following Dickson et al. (2007).

All of these calculations along with the resulting monthly $CO_2$ flux were carried out using the open source FluxEngine toolbox (Holding et al., 2019; Shutler et al., 2016), for the period between July 2002 and December 2018, assuming 'rapid' transfer (as described in Woolf et al., 2016).

## 2.3. Biological data

The 4 km resolution mean monthly chlorophyll-*a* (Chl *a*) was calculated from Moderate Resolution Imaging Spectroradiometer on Aqua (MODIS-A) Level 1 granules, retrieved from the National Aeronautics Space Administration (NASA) Ocean Colour website (https://oceancolor.gsfc.nasa.gov/; last accessed 10/12/2020), using SeaDAS v7.5, and applying the standard OC3-CI algorithm for Chl *a* (https://oceancolor.gsfc.nasa.gov/atbd/chlor_a/; last accessed 15/12/2020). Monthly composites of MODIS-A SST (NASA OBPG, 2015) and photosynthetically active radiation (PAR; NASA OBPG, 2017b) were also downloaded from the NASA Ocean Colour website. Monthly NPP composites were generated from MODIS-A Chl *a*, SST and PAR composites using the Wavelength Resolving Model (Morel, 1991) with the look up table described in Smyth et al. (2005). Coincident monthly composites of NCP using the algorithm NCP-D described in Tilstone et al. (2015) were generated using the NPP and SST data. Further details of the satellite algorithms are given in O'Reilly et al. (1998), O'Reilly and Werdell (2019) and Hu et al. (2012) for Chl *a*, Smyth et al. (2005), Tilstone et al. (2005, 2009) for NPP and Tilstone et al. (2015) for NCP. Monthly composites were generated between July 2002 and December 2018 and were re-gridded onto the same 1° grid as the $pCO_{2\,(sw)}$ and flux data. Ford et al. (2021b) showed that these satellite algorithms for Chl *a*, NPP, NCP and SST are accurate compared to *in situ* observations in the South Atlantic Ocean following an algorithm intercomparison which accounted for model, *in situ* and input parameter uncertainties.

## 2.4. Seasonal and interannual driver analysis

The X-11 analytical econometric tool (Shiskin et al., 1967) was used to decompose the timeseries into seasonal, interannual and residual components following the methodology of Pezzulli et al. (2005). In brief, the X-11 method comprises a three step filtering algorithm: (1) The interannual component ($T_t$) is initially estimated using an annual centred running mean, which is subtracted from the initial timeseries ($X_t$) to estimate the seasonal component ($S_t$). (2) $T_t$ is revised by applying an annual centred running mean to the $X_t$ minus $S_t$. The revised $T_t$ is removed from $X_t$ and the final $S_t$ calculated. (3) The final $T_t$ is calculated by applying an annual centred running mean to $X_t$ minus the revised $S_t$. The analysis has been shown to be effective in the decomposition of environmental time-series (Pezzulli et al., 2005; Vantrepotte & Mélin, 2011; Henson et al., 2018), that allows the seasonal cycle to vary on a yearly basis and, produces an interannual component that results in a robust representation of the longer-term changes in the timeseries.

The approach was applied to monthly 1° fields of $\Delta p CO_2$ that were estimated from $p CO_{2 (atm)}$ and SA-FNN $p CO_{2 (sw)}$, on a per pixel basis. The $p CO_{2 (atm)}$ and spatially and temporally varying $p CO_{2 (sw)}$ uncertainties (Table 1; Fig. B1) were propagated through the X-11 analysis, using a Monte Carlo uncertainty propagation approach. The input time series were randomly perturbed 1000 times within the uncertainties of each parameter, and Spearman correlations calculated for each perturbation. The 95% confidence interval was extracted from the resulting distribution of correlations coefficients, and results were deemed significant ($\alpha < 0.05$) where the confidence interval remained significant. Spatial autocorrelation was tested using the method of field significance (Wilks, 2006). The analysis was then conducted on the $CO_2$ fluxes, on a per pixel basis. The $p CO_{2 (sw)}$, $p CO_{2 (atm)}$, gas transfer velocity, SST and SSS uncertainties (Table 1) were propagated through the flux calculations using the same Monte Carlo uncertainty propagation approach used for $\Delta p CO_2$.

The potential drivers tested were MODIS-A skin SST, NCP and NPP alongside SSS from the CMEMS global reanalysis product (GLORYSV12; CMEMS, 2021) and two climate indices: Multivariate ENSO Index (MEI) as an indicator of El Niño Southern Oscillation phases, https://www.esrl.noaa.gov/psd/enso/mei (last accessed: 19/12/2019); Southern Annular Mode (SAM) data, which indicate the displacement of the westerly winds in the Southern Ocean, were downloaded from http://www.nerc-bas.ac.uk/icd/gjma/sam.html (last accessed: 19/12/2019).

**Table 1: Uncertainties in the input parameters used in the Monte Carlo uncertainty propagation.**

| Parameter | Uncertainty | Reference |
|---|---|---|
| $p CO_{2 (sw)}$ | Variable (Appendix B) | (Ford et al., 2022) |
| SST | 0.441 °C | (Ford et al., 2021b) |
| SSS | 0.1 psu | (Jean-Michel et al., 2021) |
| $p CO_{2 (atm)}$ | 1 µatm | (Takahashi et al., 2009) |
| Gas transfer velocity | 20 % | (Woolf et al., 2019) |

## 2.5. Trend analysis

150 The linear trend in the interannual components of $\Delta p CO_2$ and the $CO_2$ flux were calculated on a per pixel basis using the non parametric Mann-Kendall test (Kendall, 1975; Mann, 1945) and Sen's Slope estimates (Sen, 1968), which are less sensitive to outliers in the timeseries. The input parameter uncertainties (Table 1) were propagated within this trend analysis using a Monte Carlo uncertainty propagation ($n = 1000$) to extract the 95% confidence interval on the trends. The overall trend was deemed significant if 95% of the trends were significant ($\alpha = 0.05$), and the uncertainties in these trends are displayed in 155 Appendix B (Fig. B2).

## 2.6. Limitations

It should be noted that correlations between the $\Delta p CO_2$ and SST/NCP are expected since the SA-FNN estimates $p CO_{2\,(sw)}$ (the major determinant of $\Delta p CO_2$ variability) using SST and NCP as input parameters which are subsequently interpreted as drivers here. By extension, but to a lesser extent, this also applies to correlations between $CO_2$ flux and SST/NCP since $p CO_2$ 160 $_{(sw)}$ is included in the flux calculations. Different lines of evidence suggest that this is not a major limitation of our study. Firstly, any correlation between $\Delta p CO_2$/$CO_2$ flux and SST/NCP is not determined *a priori*, but is an emerging property of the SA-FNN. Therefore, the driver analysis undertaken here represents an indirect decomposition of the SA-FNN drivers rather than a strict correlation analysis between independent variables. The accurate representation of seasonal $p CO_{2\,(sw)}$ cycles across the South Atlantic Ocean (Ford et al., 2022) provides confidence in the SA-FNN. Secondly, conducting the analysis 165 described by Henson et al. (2018) using *in situ* $p CO_{2\,(sw)}$ to estimate $\Delta p CO_2$ on a per province basis (Longhurst, 1998) for the South Atlantic Ocean, yielded similar seasonal drivers to the SA-FNN (Appendix A). The interannual drivers displayed some differences however, which may be due to the spatial and temporal averaging that is required to construct the in situ timeseries.

## 3. Results

170 **3.1. Seasonal drivers of $\Delta p CO_2$ and $CO_2$ flux**

The X-11 analysis conducted on $\Delta p CO_2$ indicated significant seasonal correlations (Fig. 1), when the uncertainties are accounted for. The subtropics (10° S to 40° S) showed positive correlations between $\Delta p CO_2$, SST and SSS (Fig. 1c, d), as well as negative correlations between $\Delta p CO_2$, NCP and NPP (Fig. 1a, b). In contrast the subpolar (south of 40° S) and equatorial regions (10° N to 10° S) displayed negative correlations between $\Delta p CO_2$ and SST (Fig. 1c). Correlations between 175 $\Delta p CO_2$ and NCP were negative in the subpolar regions and were positive in the Equatorial regions (Fig. 1a). There were no significant correlations observed between $\Delta p CO_2$ and MEI or SAM in any of the regions.

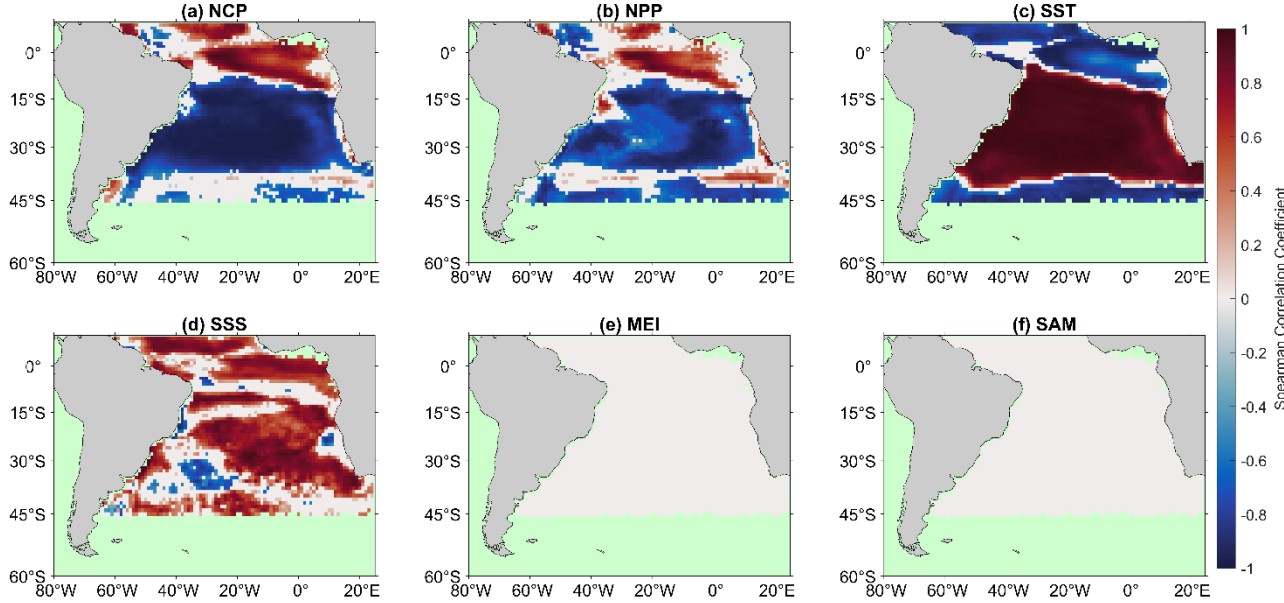

**Figure 1: Significant Spearman correlations between the $\Delta p$CO₂ seasonal component of the X-11 analysis and (a) net community production (NCP), (b) net primary production (NPP), (c) sea surface temperature (SST), (d) sea surface salinity (SSS), (e) Multivariate ENSO index (MEI) and (f) Southern Annular Mode (SAM) seasonal components. White regions indicate no significant correlations, and green regions indicate no analysis was performed due to missing satellite data.**

Regional deviations were observed in the Amazon Plume, Benguela upwelling, the South American coast, and a band across 40° S. The region under the influence of the Amazon Plume indicated negative correlations between $\Delta p$CO₂ and NCP in contrast to the surrounding waters which had positive correlations (Fig. 1a). The Benguela upwelling displayed positive correlations between $\Delta p$CO₂ and NCP (Fig. 1a), no significant correlations between $\Delta p$CO₂ and SST (Fig. 1c), and negative correlations between $\Delta p$CO₂ and SSS (Fig. 1e). The South American coast between 12° S and 17° S displayed positive correlations between $\Delta p$CO₂ and NPP (Fig. 1b), along with negative correlations between $\Delta p$CO₂ and SSS (Fig. 1e). Negative correlation between $\Delta p$CO₂ and SSS, and positive correlations between NCP, NPP and $\Delta p$CO₂ were also observed in the southwestern Atlantic (Fig. 1e). Positive correlations between NCP, NPP and $\Delta p$CO₂ were identified in a band across 40° S (Fig. 1a, b). Performing the X-11 analysis on the CO₂ flux revealed similar and comparable correlations to $\Delta p$CO₂ (Fig. 2). Significant driver-flux correlations were observed over a larger area however, compared to $\Delta p$CO₂.

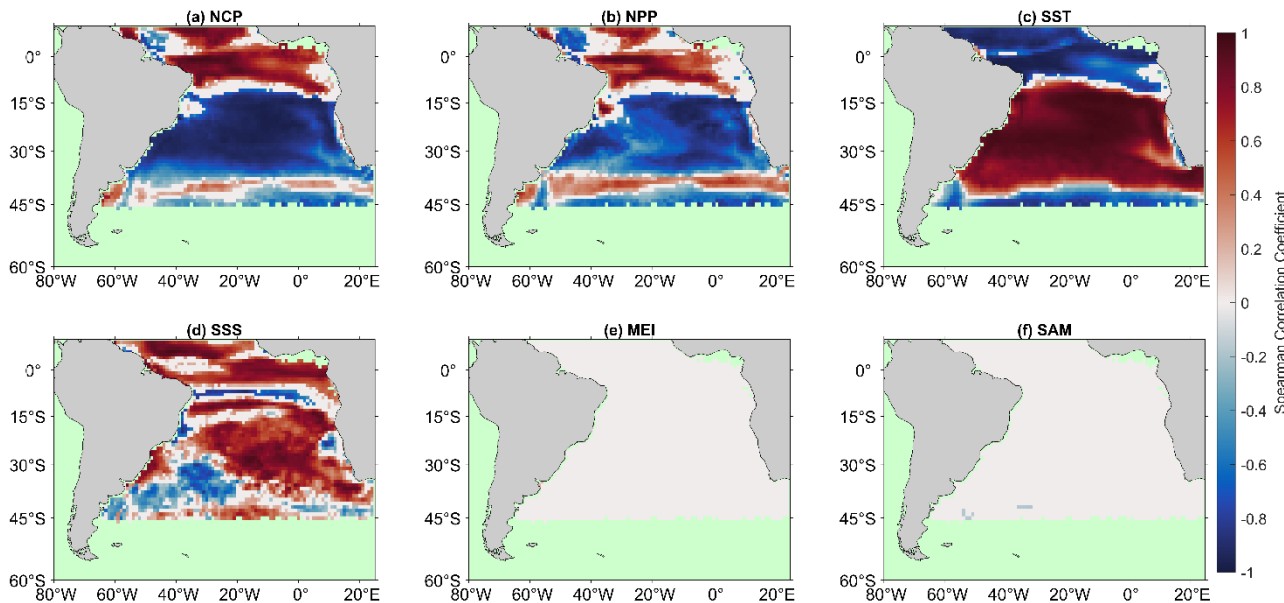

Figure 2: Significant Spearman correlations between the air-sea $CO_2$ flux seasonal component of the X-11 analysis and (a) net community production (NCP), (b) net primary production (NPP), (c) sea surface temperature (SST), (d) sea surface salinity (SSS), (e) Multivariate ENSO index (MEI) and (f) Southern Annular Mode (SAM) seasonal components. White regions indicate no significant correlations, and green regions indicate no analysis was performed due to missing satellite data.

## 3.2. Interannual drivers of $\Delta p CO_2$ and $CO_2$ flux

The X-11 analysis identified regionally significant interannual correlations between $\Delta p CO_2$ and SST, MEI and to a lesser extent NCP and SSS (Fig. 3). The subtropics displayed positive correlations between SST and $\Delta p CO_2$, which extended across the basin from the South American coast (Fig. 3c). Positive correlations were also observed between the MEI and $\Delta p CO_2$ (Fig. 3e), with a similar geographic extent as the correlations with SST. In the central South Atlantic gyre spatially variable negative correlations between NCP and $\Delta p CO_2$, and positive correlations between SSS and $\Delta p CO_2$ were observed (Fig. 3a, d). The central Equatorial Atlantic displayed spatially variable positive correlations between NCP and $\Delta p CO_2$, which extended south-east towards the African coast (Fig. 3a).

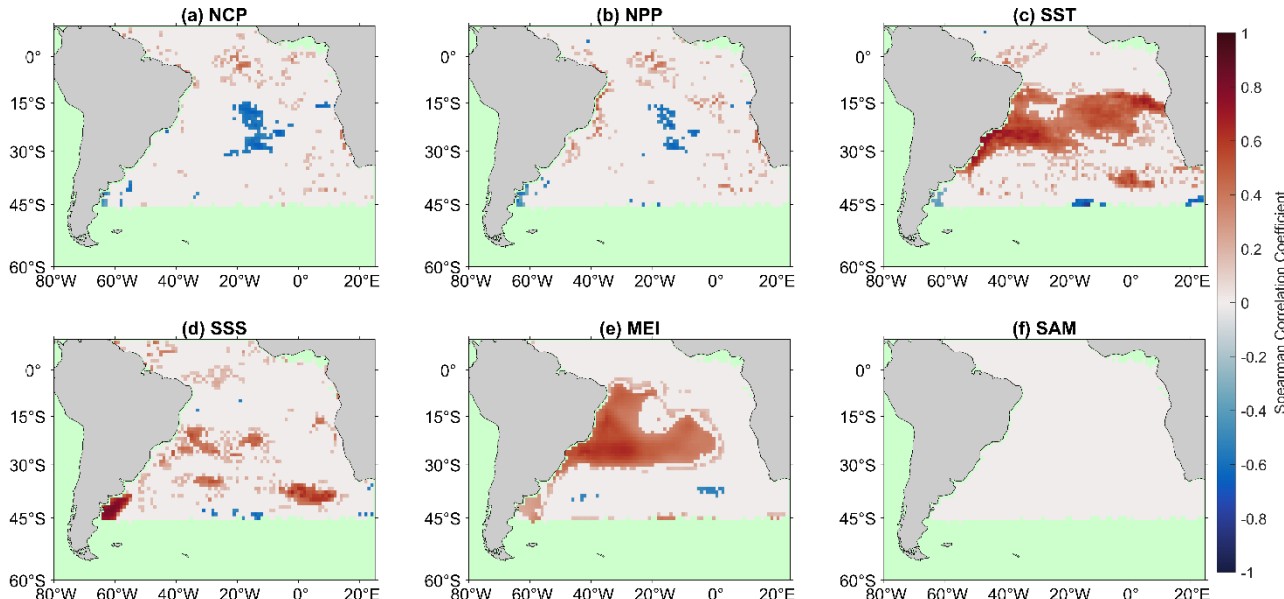

**Figure 3: Significant Spearman correlations between the $\Delta p\mathrm{CO_2}$ interannual component of the X-11 analysis and (a) net community production (NCP), (b) net primary production (NPP), (c) sea surface temperature (SST), (d) sea surface salinity (SSS), (e) Multivariate ENSO index (MEI) and (f) Southern Annular Mode (SAM) interannual components. White regions indicate no significant correlations, and green regions indicate no analysis was performed due to missing satellite data.**

Significant interannual correlations for the $\mathrm{CO_2}$ flux were also identified by the X-11 analysis (Fig. 4), which generally covered a larger spatial area to the corresponding $\Delta p\mathrm{CO_2}$ correlations (Fig. 3). Positive correlations between the $\mathrm{CO_2}$ flux and SST were observed in the subtropics (Fig. 4c), consistent with the correlations with $\Delta p\mathrm{CO_2}$ (i.e. by comparing Fig. 4c and Fig. 3c). Nevertheless, negative correlations between the $\mathrm{CO_2}$ flux and SST were observed at the border between the equatorial region and subtropics, which was not identified in the $\Delta p\mathrm{CO_2}$ correlations. Negative correlations between NCP and the $\mathrm{CO_2}$ flux were also identified over a spatially larger area (Fig. 4a, 3a). Correlations between the MEI and $\mathrm{CO_2}$ flux were positive in the subtropics (Fig. 4e) and included a band of negative correlations to the south between 35° S and 45° S (Fig. 4e).

Positive correlations between NCP and $\mathrm{CO_2}$ flux were observed in the western equatorial Atlantic, alongside spatially variable negative correlations to SST (Fig. 4a, c). Positive correlations between SSS and $\mathrm{CO_2}$ flux were identified in the region of the Amazon plume (Fig. 4d). Weak positive correlations between the SAM and $\mathrm{CO_2}$ flux were identified between 30° S and 45° S (Fig. 4f).

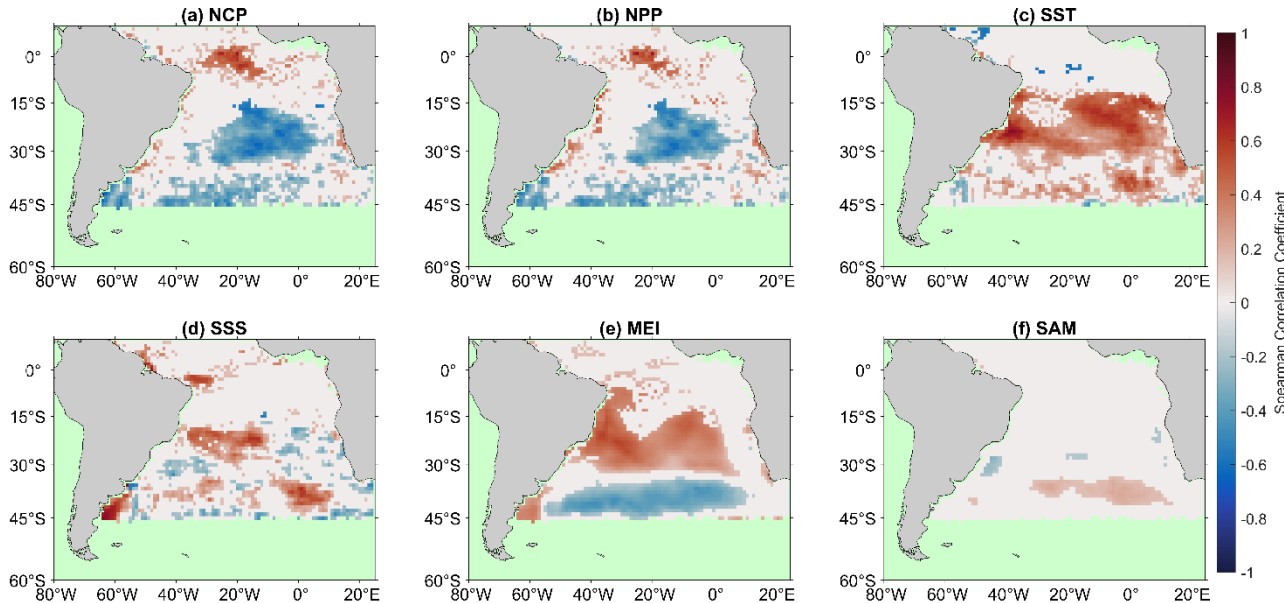

**Figure 4: Significant Spearman correlations between the air-sea CO$_2$ flux interannual component of the X-11 analysis and (a) net community production (NCP), (b) net primary production (NPP), (c) sea surface temperature (SST), (d) sea surface salinity (SSS), (e) Multivariate ENSO index (MEI) and (f) Southern Annular Mode (SAM) interannual components. White regions indicate no significant correlations, and green regions indicate no analysis was performed due to missing satellite data.**

### 3.3. Trends in interannual $\Delta p$CO$_2$ and CO$_2$ flux

Regions of significant trends in the interannual component of $\Delta p$CO$_2$ were observed (Fig. 5a). Negative trends occurred in the South Atlantic gyre. Positive trends in $\Delta p$CO$_2$ were identified along the South African coast, which switched to strong negative trends moving offshore into the central South Atlantic gyre. Positive trends were also observed in the Equatorial Atlantic consistent with the positions of the Amazon Plume and Equatorial Upwelling.

Regions of significant trends in the CO$_2$ flux were identified (Fig. 5b), but over much larger spatial areas than evident in the $\Delta p$CO$_2$ results (i.e. comparing Fig. 5a with 5b). The trends in CO$_2$ flux are generally in the same direction as trends in $\Delta p$CO$_2$. Strong positive trends in the CO$_2$ flux occurred in the Benguela upwelling region, before switching to a negative trend offshore of similar magnitude but occupying a larger spatial extent.

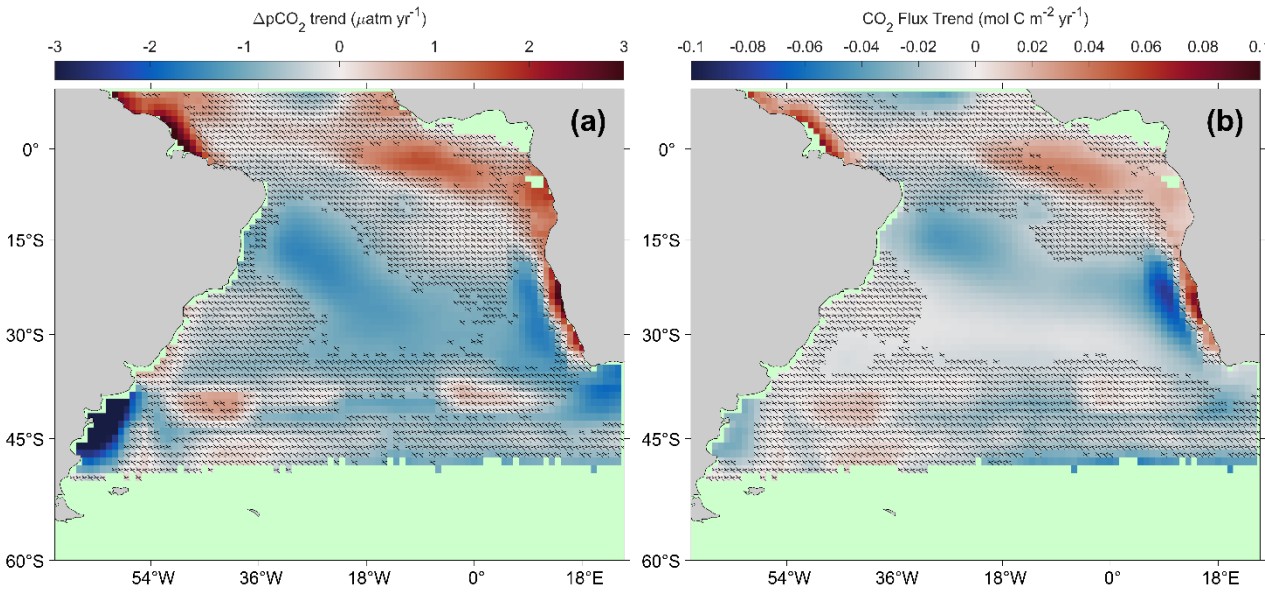

240

**Figure 5: Linear trends in (a) $\Delta p$CO$_2$ and (b) the air-sea CO$_2$ flux between 2002 and 2018. Hashed areas indicate non-significant trends when accounting for the uncertainties. Green regions indicate insufficient data to calculate trends.**

## 4. Discussion

### 4.1. Seasonal drivers of $\Delta p$CO$_2$ and CO$_2$ flux

245 Previous studies have explored the seasonal drivers of $\Delta p$CO$_2$ and to a lesser extent the air-sea CO$_2$ flux. In this study, we investigated the drivers of $\Delta p$CO$_2$ and CO$_2$ flux at both seasonal and interannual timescales in the South Atlantic Ocean. In the North Atlantic, Henson et al. (2018) indicated that the seasonal variability in subtropical $\Delta p$CO$_2$ variability is driven by SST, whereas the variability in $\Delta p$CO$_2$ in subpolar regions is biologically driven, similar to previous studies (Takahashi et al., 2002; Landschützer et al., 2013). The X-11 analysis conducted on spatially complete $\Delta p$CO$_2$ and CO$_2$ flux displayed

250 consistent seasonal results (Fig. 1, 2), though for the CO$_2$ flux significant correlations occupied a larger area. These both indicated a similar pattern in seasonal drivers for the South Atlantic Ocean, with subtropical $\Delta p$CO$_2$ and CO$_2$ flux driven by SST, and subpolar correlated with biological controls, although the equatorial region exhibited more complex patterns (Fig. 1).

In the Equatorial Atlantic, the correlations between $\Delta p$CO$_2$, SST and biological production were spatially variable (Fig. 1).

255 Landschützer et al. (2013) suggested that the temperature and non-temperature (i.e. biological and circulation) drivers generally compensated each other. We found positive correlations between the NCP, $\Delta p$CO$_2$ and CO$_2$ flux seasonal components, indicating that biological activity is likely a key driver of seasonal variability in response to the equatorial

upwelling. Ford et al. (2022) showed that the SA-FNN improved the seasonal $pCO_{2\,(sw)}$ variability in the Equatorial Atlantic compared to the current 'state of the art' SOM-FNN methodology (Watson et al., 2020). Elevated $\Delta pCO_2$ associated with elevated NCP in the eastern Equatorial Atlantic was consistent with the seasonal equatorial upwelling (Radenac et al., 2020). Parard et al. (2010) indicated strong negative correlations between SST and $\Delta pCO_2$ during the upwelling season (R= -0.76 for June to September), which is also consistent with our results. By contrast, Lefèvre et al. (2016) showed that correlations between $pCO_{2\,(sw)}$ and SST were weak across the whole year (R= -0.13), and SSS (R = 0.93) was the primary driver at the same station.

In the western Equatorial Atlantic, negative correlations between NCP and $\Delta pCO_2$, and positive correlations between the SSS and $\Delta pCO_2$ seasonal component occurred in the vicinity of the Amazon River mouth. The mixing of the Amazon river and oceanic water decreases SSS (Ibánhez et al., 2016; Lefèvre et al., 2020; Bonou et al., 2016; Lefévre et al., 2010), and increases the nutrient supply to the ocean which can in turn enhance NPP and NCP, leading to a decrease in $\Delta pCO_2$ within the Amazon plume (Körtzinger, 2003; Cooley et al., 2007). This coupling produces an extensive area of depressed $\Delta pCO_2$ which is a $CO_2$ sink (Ibánhez et al., 2016). Lefèvre et al. (2010) indicated that rainfall from the intertropical convergence zone could reduce SSS, with an associated decrease in $\Delta pCO_2$. The Eastern Tropical Atlantic is also subject to large river input, especially from the Congo (Hopkins et al., 2013) and Niger rivers, which could produce nutrient-rich plumes that fuel NCP and decrease $\Delta pCO_2$ (Lefèvre et al., 2016, 2021).

Between 30° S and 45° S, dissolved inorganic carbon and SST exert a similar influence on $pCO_{2\,(sw)}$, indicating that seasonal changes in dissolved inorganic carbon driven by biological uptake in the summer and upwelling in winter are approximately balanced by seasonal changes in SST and their control on the solubility pump (Henley et al., 2020). This likely explains the band of positive correlations between NCP, NPP and $\Delta pCO_2$ and sharp transitions in correlations between SST and $\Delta pCO_2$ across ~40° S.

Deviations from the expected drivers in the subtropics, occurred within the Benguela upwelling system between 20° S and 35° S. Positive correlations between NCP and the $CO_2$ flux (Fig. 2a) alongside negative correlations between SST, SSS and the $CO_2$ flux (Fig. 2c, d) are indicative of upwelled waters that have both elevated $pCO_{2\,(sw)}$ and nutrients, which cause an increase in NPP (Lamont et al., 2014). These upwelled waters move offshore in filaments (Rubio et al., 2009) where NPP decreases, and SST becomes the dominant driver, which is confirmed by the positive correlations between SST and the $CO_2$ flux further offshore. Ford et al. (2021b) indicated a switch in NCP drivers in the Benguela upwelling from wind driven upwelling on the shelf, to filaments that propagate offshore from the upwelling front, which is consistent with the switch in the drivers observed for the $CO_2$ flux as these filaments move offshore.

At between 12° S and 17 °S along the South American coast, there were also deviations from the expected drivers as there were positive correlations between NPP and $\Delta pCO_2$ (Fig. 1b) and negative correlations between SSS and $\Delta pCO_2$ (Fig. 1d), which are consistent with an upwelling signature that occurs along the coast. Aguiar et al. (2018) also showed intense seasonal upwelling events in this region that are driven by wind and currents. The southern coast of South America is strongly influenced by riverine water input that reduces the total alkalinity and therefore causes an increase in $pCO_{2\,(sw)}$

(Liutti et al., 2021). This is associated with an increased supply of nutrients which in turn enhances NPP, though the main drivers of $p\mathrm{CO}_{2\,(sw)}$ in this region still remain as total alkalinity and SST (Liutti et al., 2021). This potentially explains the positive correlation between $\Delta p\mathrm{CO}_2$ and both NCP and NPP (Fig. 1a, b), as well as the negative correlations between $\Delta p\mathrm{CO}_2$
and SSS. The extension offshore of this negative correlation between SSS and $\Delta p\mathrm{CO}_2$ (Fig. 1d) could be caused by the advection of water masses due to intense mesoscale eddy activity arising from the Brazil-Malvinas confluence (Mason et al., 2017).

The seasonal correlations between the $\mathrm{CO}_2$ flux and the drivers were similar to $\Delta p\mathrm{CO}_2$, but for $\mathrm{CO}_2$ flux these occurred over a larger spatial area. The South Atlantic subtropical anticyclone (Reboita et al., 2019) which controls wind speeds across the
region, and therefore the gas transfer velocity could enhance the $\mathrm{CO}_2$ flux into the subtropical ocean, through higher (or lower) wind speeds in winter (or summer; Xiong et al., 2015). Since seasonal variations in $\Delta p\mathrm{CO}_2$ largely explain the seasonal variability in the $\mathrm{CO}_2$ flux $\Delta p\mathrm{CO}_2$ can be used as a proxy to understand seasonal variations in the $\mathrm{CO}_2$ flux in this region.

### 4.2. Interannual drivers of $\Delta p\mathrm{CO}_2$ and $\mathrm{CO}_2$ flux

The larger geographic region of significant correlations for the air-sea $\mathrm{CO}_2$ flux compared to $\Delta p\mathrm{CO}_2$, and the consistency between the two results (i.e. comparing the smaller regions of $\Delta p\mathrm{CO}_2$ correlations with their equivalent in the flux results; Fig. 3, 4) suggests that analysing the $\mathrm{CO}_2$ flux is the better dataset to investigate drivers of variations in interannual and longer timescales. The results become clearer when analysing the $\mathrm{CO}_2$ flux, where the effects of solubility and gas transfer (estimated via wind speed proxy) could reinforce correlations and multi-year trends, which will be retrieved by performing
long timeseries analyses on the $\mathrm{CO}_2$ flux. Landschützer et al. (2015) showed that variations in the Southern Ocean carbon sink were primarily driven by changes in $\Delta p\mathrm{CO}_2$, when integrating across basin scales. At localised scales of $1°$ by $1°$ as performed in our analysis, changes in surface turbulence and solubility are shown to be important in determining interannual variability, consistent with Keppler and Landschützer (2019). In the North Atlantic Ocean, Henson et al. (2018) showed that the seasonal and interannual drivers of $\Delta p\mathrm{CO}_2$ are different, which could arise from the necessity to study $\mathrm{CO}_2$ fluxes over
longer timescales.

The interannual component of NCP and the $\mathrm{CO}_2$ flux were negatively correlated in the subtropical gyre (Fig. 4a), alongside a positive correlation between SST and $\mathrm{CO}_2$ flux (Fig. 4b). El Niño (La Niña) events are known to influence the South Atlantic Ocean, causing an increase (decrease) in SST across the basin (Rodrigues et al., 2015; Colberg et al., 2004), and a decrease (increase) in NPP and NCP (Ford et al., 2021b; Tilstone et al., 2015). Positive correlations between the MEI and $\mathrm{CO}_2$ flux
(Fig. 4e) indicate that the MEI partially controls the interannual variability in $\mathrm{CO}_2$ flux in the South Atlantic subtropical gyre, through modulations primarily in SST and to a lesser extent NCP. The South Atlantic Subtropical Anticyclone has been observed to strengthen (weaken) and move south (north) during La Niña (El Niño) events. This displacement increases (decreases) wind speeds across the subtropical South Atlantic, which will enhance (weaken) gas exchange, and elevate

(depress) NCP (Ford et al., 2021b). These results suggest a more significant role of NCP in controlling the interannual
variability in the $CO_2$ flux than has previously been thought.

The negative correlation between the $CO_2$ flux and the MEI in a band between 30° S and 45° S (Fig. 4e), indicates that reduced (elevated) wind speeds that occur during La Niña (El Niño) events in this region, suppress (enhance) the gas exchange (Colberg et al., 2004) and therefore acts as a weaker (stronger) $CO_2$ sink. In the equatorial region, neither $\Delta p CO_2$ or the $CO_2$ flux were correlated with the MEI, in sharp contrast with Lefèvre et al. (2013) who showed stronger outgassing of
$CO_2$ in the western equatorial Atlantic for the year following the 2009 El Niño. In that respect, it should be noted that our analysis would not identify such lagged correlations.

The SAM has known meteorological connections to the MEI (Fogt et al., 2011), where El Niño (La Niña) events generally coincide with negative (positive) SAM phases, resulting in northward (southward) displacement of the westerly winds in the Southern Ocean. Our results showed positive correlations between the $CO_2$ flux and the SAM between 30° S and 45° S (Fig.
4f) indicating stronger (weaker) $CO_2$ drawdown into the oceans during negative (positive) SAM phases. Although no significant correlations were found between $\Delta p CO_2$ and the SAM (Fig. 3f), the changes in the gas transfer driven by the displacement of the westerly winds could control the $CO_2$ flux. It should be noted that the effect of the SAM may be more pronounced outside the domain of the present study (i.e south of 45 °S; Keppler and Landschützer, 2019). Landschützer et al. (2015) indicated that the SAM is unlikely to be the main driver of changes in the Southern Ocean $CO_2$ flux, but an
observed zonally asymmetric atmospheric pattern could induce changes in the $CO_2$ flux (Keppler and Landschützer, 2019; Landschützer et al., 2015). This asymmetric atmospheric pattern, however, may not be captured within the SAM index.

### 4.3. Multi-year trends in $\Delta p CO_2$ and $CO_2$ flux

The trends in $\Delta p CO_2$ and $CO_2$ flux over 16 years (Fig. 5) showed some similarities to previous trend assessments in the South Atlantic Ocean (Landschützer et al., 2016). Our results indicated a lower number of significant trends however, since
uncertainties in the trend analysis were accounted for. The uncertainties in both the $pCO_{2\ (sw)}$ estimates from extrapolation techniques and the gas transfer velocity are rarely propagated through previous trend analyses. By accounting for these uncertainties, the trend analyses provide a robust depiction of regions that can confidently be determined as changing. As with the seasonal and interannual analysis, the $CO_2$ flux-based trend analysis showed a greater spatial area of significant trends, when compared to $\Delta p CO_2$ (Fig. 5).
The strongest trends in $\Delta p CO_2$ and the $CO_2$ flux were observed in the Benguela upwelling system. Arnone et al. (2017) reported positive trends in *in situ* $pCO_{2\ (sw)}$ of 6.1 ± 1.4 µatm yr$^{-1}$, between 2005 and 2015. Assuming an atmospheric $CO_2$ increase of 1.5 µatm yr$^{-1}$ (Takahashi et al., 2002; Zeng et al., 2014), these results are consistent with the $\Delta p CO_2$ trends observed in this study (1.5 – 3.8 µatm yr$^{-1}$, Fig. 5a). Arnone et al. (2017) also suggested that the positive trend was due to a stronger influence of upwelling (Rouault et al., 2010), which injects $CO_2$ and nutrients into the area that is then not
completely removed by the enhanced NPP/NCP. Varela et al. (2015) indicated an increase in the strength of the Benguela upwelling. By contrast, Lamont et al. (2018) showed no significant change in upwelling in the Southern Benguela but

increases in the Northern Benguela which are consistent with our data that highlights an increasing efflux of $CO_2$ to the atmosphere (Fig. 5b). The $CO_2$ flux trends in this study (0.03 – 0.09 mol m$^{-2}$ yr$^{-1}$, Fig. 5b) were also consistent with but slightly lower than the 0.13 ± 0.03 mol m$^{-2}$ yr$^{-1}$ trend in $CO_2$ flux observed by Arnone et al. (2017). An increase in the strength of the upwelling that injects $CO_2$ into the surface layer, will be driven by enhanced (upwelling-conducive) winds, that also enhance the gas transfer. This highlights the importance of studying multi-year trends using the $CO_2$ flux, because the enhancement of these trends by meteorological conditions would not be observed using $\Delta pCO_2$ alone.

Offshore from the upwelling region negative $\Delta pCO_2$ and $CO_2$ flux trends were observed. Rubio et al. (2009) showed that mesoscale filaments and eddies propagate away from the upwelling front, transporting nutrients offshore into the South Atlantic gyre. Ford et al. (2021b) showed negative correlations between sea level height anomalies (SLHA), and NPP/NCP anomalies (negative SLHA; positive NCP/PPP), indicating an influence of mesoscale features on $\Delta pCO_2$ and the $CO_2$ flux. Xiu et al. (2018) indicated that an increase in upwelling conducive winds could increase the number of mesoscale eddies, which would transport nutrients offshore of the Californian upwelling. Although the Benguela and Californian upwelling systems are not identical, these connections could suggest an elevated nutrient export offshore, driving elevated NPP/NCP, which would increase the $CO_2$ sink. Kulk et al. (2020) showed significant increases in NPP of ~2 % yr$^{-1}$, between 1998 and 2018 in the region of strong negative trends in the $CO_2$ flux observed in this study, which supports the  contribution of NCP to multi-year trends in the $CO_2$ flux.

There were also positive trends in $\Delta pCO_2$ and $CO_2$ flux in the Equatorial Atlantic. In the Eastern Equatorial Atlantic, Lefèvre et al. (2016) previously suggested a negative trend in *in situ* $\Delta pCO_2$, between 2006 and 2013, but indicated that the trend may be biased by extreme events at either end of the record. From 1995 to 2007, Parard et al. (2010) indicated a greater increase in *in situ* $pCO_{2\,(sw)}$ than $pCO_{2\,(atm)}$ (increasing $\Delta pCO_2$), but the trend was derived from data from only two research cruises. For the Equatorial upwelling, an increase in $\Delta pCO_2$ (as shown here and in Landschützer et al., 2016) is counter intuitive because there is evidence that upwelled water has recently been in contact with the atmosphere (~15 years; Reverdin et al., 1993). Dissolved inorganic carbon in these upwelled waters has been shown to increase at a similar rate to the surface waters (e.g Woosley et al., 2016). Therefore, the trend in $\Delta pCO_2$ should be ~0 with increasing $pCO_{2\,(atm)}$. This could suggest a missing component within the SA-FNN to estimate $pCO_{2\,(sw)}$, such as changes in the biological export efficiency (Kim et al., 2019), which could then suppress upwelling induced $CO_2$ outgassing.

The Western Tropical Atlantic, in the vicinity of the Amazon Plume, also showed positive trends in $\Delta pCO_2$ and $CO_2$ flux. Previous studies have not investigated the trends in $\Delta pCO_2$ or $CO_2$ flux in the Amazon Plume, however the carbon retention in a colored ocean site (CARIACO), situated to the northwest, displayed positive trends in $pCO_{2\,(sw)}$ of 2.95 ± 0.43 µatm yr$^{-1}$ (Bates et al., 2014). Araujo et al. (2019) identified a positive trend in $pCO_{2\,(sw)}$ of 1.20 µatm yr$^{-1}$, but a trend in $pCO_{2\,(atm)}$ of 1.70 µatm yr$^{-1}$ (i.e. decreasing $\Delta pCO_2$) for the northeast Brazilian coast, Although, the air-sea $CO_2$ flux and $\Delta pCO_2$ within the Amazon Plume region is spatially and temporally variable (Valerio et al., 2021; Ibánhez et al., 2016; Bruto et al., 2017).

The South Atlantic gyre exhibited negative trends in $\Delta pCO_2$ and the $CO_2$ flux indicating an increasing drawdown of atmospheric $CO_2$ into the ocean, which were consistent with Landschützer et al. (2016) over the period from 1982 and 2011

though the trends were at the limits of the uncertainties (Fig. B2). Fay and Mckinley (2013) showed weak negative trends in $\Delta p CO_2$ using *in situ* observations over different time series lengths. Using an ensemble of complete $p CO_{2\,(sw)}$ fields, Gregor et al. (2019) indicated negative trends in $\Delta p CO_2$ however there was low confidence in these trends especially in the South Atlantic gyre. By contrast, Kitidis et al. (2017) reported a mean trend in *in situ* $\Delta p CO_2$ between 1995 and 2013, that was not

significantly different from zero. These contradictory trends support the conclusion that $\Delta p CO_2$ is unlikely to be representative of the $CO_2$ flux over multi-year timescales. Therefore, we recommend that the $CO_2$ flux should be used to assess multi-year variability in the oceanic $CO_2$ sink, as the importance of changes in solubility and gas transfer velocity (estimated via wind speed) increases (Keppler and Landschützer, 2019).

During the United Nations decade of ocean science (2021-2030) , the Integrated Ocean Carbon Research (IOC-R) highlights

that the role of biology is a key issue to understanding the global ocean $CO_2$ sink (Aricò et al., 2021). The biological contribution to both interannual and multi-year variations in the South Atlantic air-sea $CO_2$ flux shown in this study, and supported by Ford et al. (2022), indicates that the biology activity through NCP cannot be assumed to be in steady state. The biological effect of NCP on $\Delta p CO_2$ and $CO_2$ flux should therefore not be overlooked when assessing the interannual and multi-year variations in the global ocean carbon sink.

**5. Conclusions**

In this paper, we have identified the seasonal and interannual drivers of $\Delta p CO_2$ and the air-sea $CO_2$ flux in the South Atlantic Ocean using satellite observations. Seasonally, our results indicated that the subtropics were controlled by SST, and the subpolar regions were correlated with biological processes. Deviations from this trend occurred in the Benguela upwelling where predominately biological processes correlated with variability in the $\Delta p CO_2$ as well as upwelling. The Equatorial

Atlantic showed spatially variable drivers associated with the Amazon Plume and Equatorial upwelling which induced a biological effect. These regions imply a strong biological control on $\Delta p CO_2$ through local physical processes. The $CO_2$ flux had similar seasonal drivers to $\Delta p CO_2$, but with significant correlations over a larger spatial area. This highlights that $\Delta p CO_2$ can be used to indicate the important drivers of the $CO_2$ flux on seasonal timescales, but it is still possible that $\Delta p CO_2$ will miss some of the spatial correlations and will likely overestimate the strength of these correlations.

The interannual variability of $\Delta p CO_2$ and the $CO_2$ flux was correlated with the MEI through a reduction (increase) of NCP and increase (decrease) in SST during El Niño (La Niña) events, again highlighting the importance of biology to the interannual variability. The $CO_2$ flux response extended over a larger geographical region, indicating that the $CO_2$ flux should be used to assess interannual trends in the oceanic $CO_2$ sink, as opposed to a proxy such as $\Delta p CO_2$, which may overestimate the strength of the correlations and does not include variability in the solubility and the gas transfer velocity

(estimated via wind speed). The 16 year trends in $\Delta p CO_2$ and the $CO_2$ flux were determined with associated uncertainties which identified negative trends in the $CO_2$ flux in the South Atlantic gyre. Positive trends in the $CO_2$ flux were observed in the Benguela upwelling region, which were associated with an increase in the strength and frequency of upwelling. A

transition to negative trends offshore were consistent with elevated nutrient export from the upwelling area, and subsequent biological drawdown of $CO_2$. These results highlight, that changes in biological activity in the South Atlantic Ocean can
control the interannual and multi-year trends in the oceanic $CO_2$ flux. This emphasises the importance of biology and specifically NCP in assessing the global ocean carbon sink.

**Appendices**

**Appendix A – Driver analysis using *in situ* $\Delta p\text{CO}_2$**

Henson et al. (2018) performed the X-11 analysis using *in situ* $p\text{CO}_{2\,(sw)}$ observations to estimate average $\Delta p\text{CO}_2$ for the
Longhurst provinces (Longhurst, 1998). The in situ $p\text{CO}_{2\,(sw)}$ observations were obtained from SOCATv2020 (https://www.socat.info/; Bakker et al., 2016), and were reanalysed to a temperature dataset representative for a consistent and fixed depth (Reynolds et al., 2002) which is used to represent the base of the mass boundary layer. The reanalysis method used the 'fe_reanalyse_socat.py' routine within FluxEngine (Holding et al., 2019; Shutler et al., 2016), which follows the methodology of Goddijn-Murphy et al. (2015), and as used in Woolf et al. (2019) and Watson et al (2020).
$\Delta p\text{CO}_2$ was calculated using the reanalysed *in situ* $p\text{CO}_{2\,(sw)}$ observations and $p\text{CO}_{2\,(atm)}$. These $\Delta p\text{CO}_2$ estimates were used within the driver analysis as described by Henson et al. (2018), using the drivers described in section 2.4, for the South Atlantic Longhurst provinces (Longhurst, 1998). The seasonal drivers of *in situ* $\Delta p\text{CO}_2$ (Fig. A1) showed a similar spatial distribution as the SA-FNN $\Delta p\text{CO}_2$ (Fig. 1). The interannual drivers (Fig. A2) showed some differences to the SA-FNN (Fig. 3). The averaging required to produce the *in situ* $\Delta p\text{CO}_2$ timeseries may mask interannual signals, and Ford et al. (2021b)
indicated that averaging over large province areas could mask correlations, especially in dynamic regions, and locally these correlations may be significant.

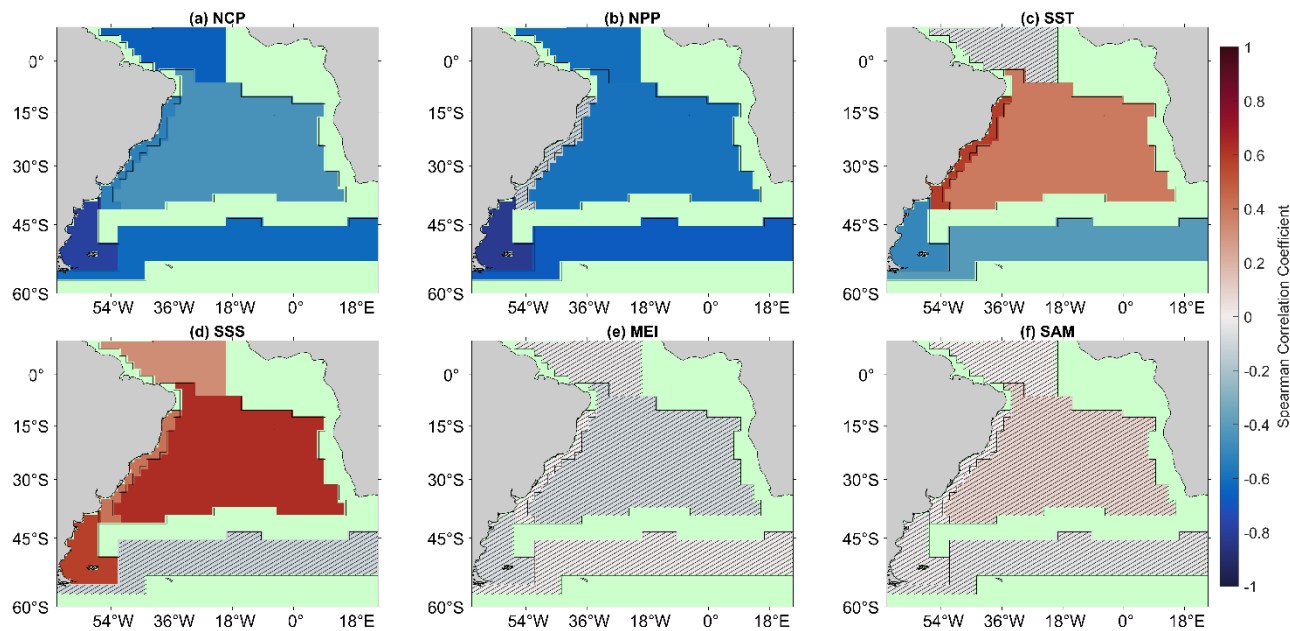

**Figure A1 - Spearman correlations between the *in situ* $\Delta p$CO$_2$ seasonal component of the X-11 analysis and (a) net community production (NCP), (b) net primary production (NPP), (c) sea surface temperature (SST), (d) sea surface salinity (SSS), (e) Multivariate ENSO index (MEI) and (f) Southern Annular Mode (SAM) seasonal components on a per province basis. Hashed areas indicate no significant correlations, and green regions indicate no analysis was performed due to missing data.**

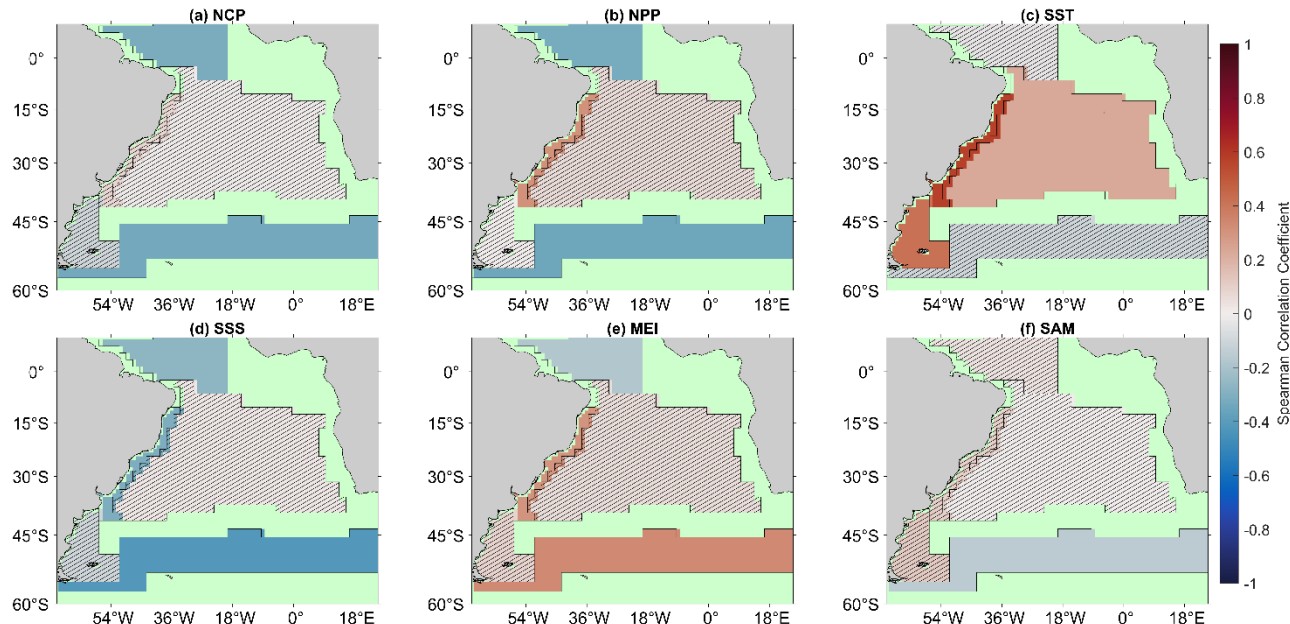

**Figure A2 - Spearman correlations between the *in situ* $\Delta p$CO$_2$ interannual component of the X-11 analysis and (a) net community production (NCP), (b) net primary production (NPP), (c) sea surface temperature (SST), (d) sea surface salinity (SSS) (e) Multivariate ENSO index (MEI) (f) Southern Annular Mode (SAM) interannual components on a per province basis. Hashed areas indicate no significant correlations, and green regions indicate no analysis was performed due to missing data.**

## Appendix B – SA-FNN $pCO_{2\,(sw)}$ and trend uncertainties

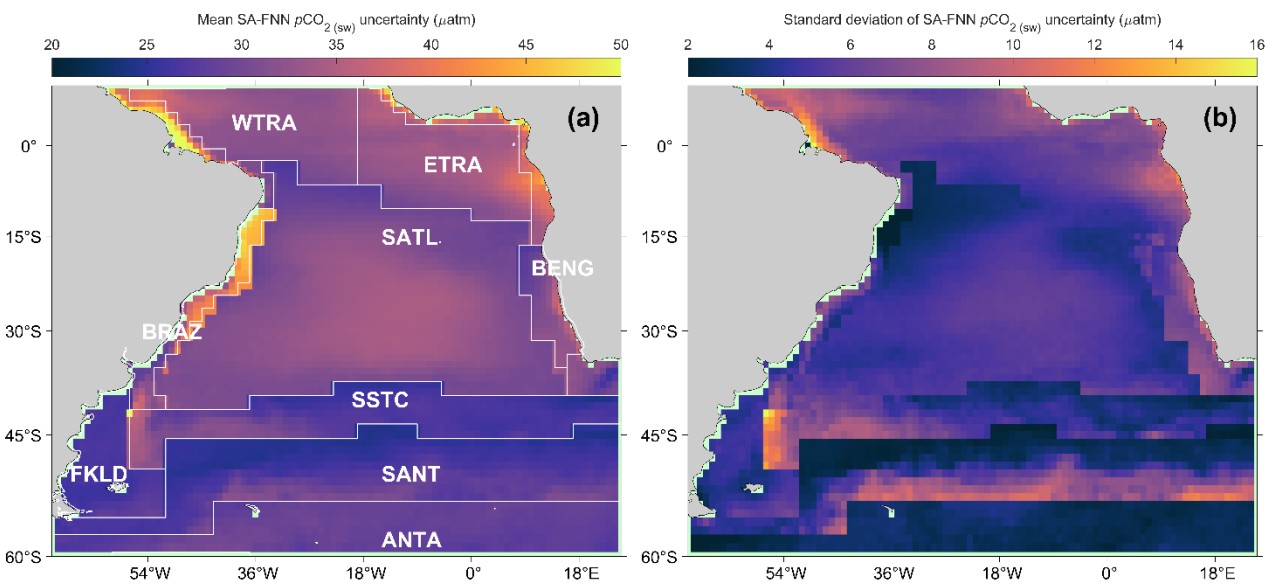

**Figure B1 – (a) Mean SA-FNN $pCO_{2\,(sw)}$ uncertainty between July 2002 and December 2018. Longhurst provinces (Longhurst, 1998) used within the SA-FNN training described in Ford et al. (2022; note the WTRA and ETRA are merged into one province). The province areas acronyms are listed as follows: WTRA is western tropical Atlantic; ETRA is eastern equatorial Atlantic; SATL is South Atlantic Gyre; BRAZ is Brazilian current coastal; BENG is Benguela Current coastal upwelling; FKLD is Southwest Atlantic shelves; SSTC is South Subtropical Convergence; SANT is sub-Antarctic and ANTA is Antarctic. (b) Standard deviation of SA-FNN $pCO_{2\,(sw)}$ uncertainty.**


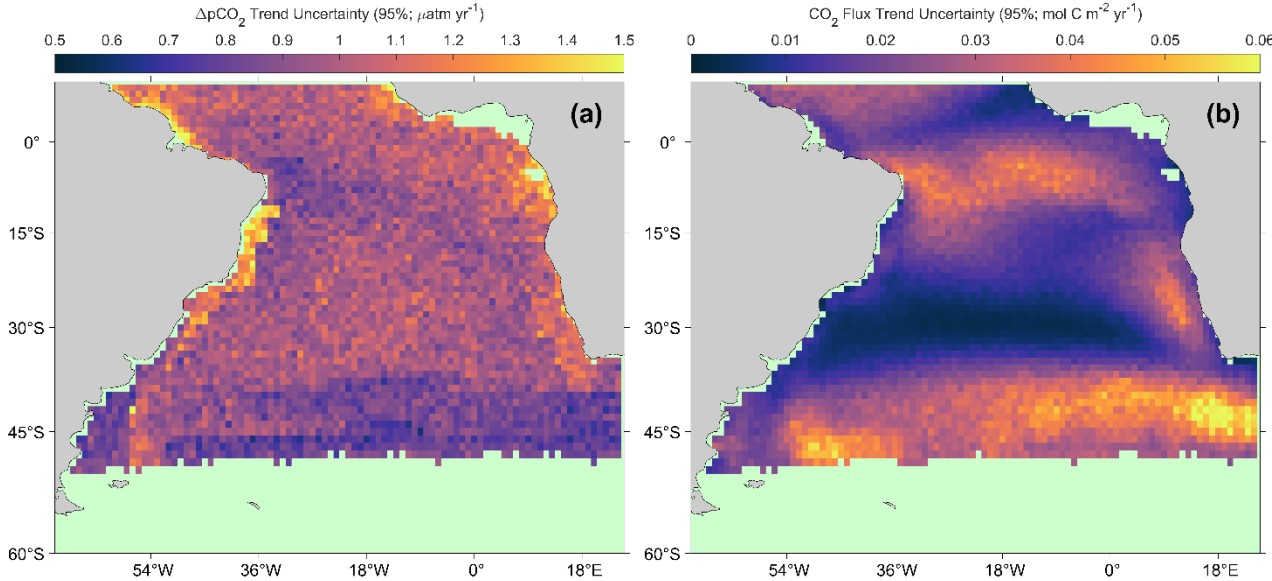


**Figure B2 – (a) Uncertainty in the ΔpCO₂ trends presented in Fig. 5a (b) Uncertainty in the air-sea CO₂ flux trends presented in Fig. 5b**

## Data Availability

Moderate Resolution Imaging Spectroradiometer on Aqua (MODIS-A) estimates of chlorophyll-*a* (NASA OBPG, 2017a),
photosynthetically active radiation (NASA OBPG, 2017b) and sea surface temperature (NASA OBPG, 2015) are available
from the National Aeronautics Space Administration (NASA) ocean colour website (https://oceancolor.gsfc.nasa.gov/).
Modelled sea surface salinity from the Copernicus Marine Environment Modelling Service global ocean physics reanalysis
product (GLORYS12V1) are available from CMEMS (CMEMS, 2021). ERA5 monthly reanalysis wind speeds are available
from the Copernicus Climate Data Store (Hersbach et al., 2019). $pCO_{2\ (atm)}$ data are available from v5.5 of the global
estimates of $pCO_{2\ (sw)}$ dataset (Landschützer et al., 2017, 2016). $pCO_{2\ (sw)}$ estimates generated by the SA-FNN are available
from Pangaea (Ford et al., 2021a). SOCATv2020 *in situ* $pCO_{2\ (sw)}$ observations (Bakker et al., 2016) are available from
https://www.socat.info/index.php/version-2020/.

## Author Contributions

DJF, GHT, JDS and VK conceived and directed the research. DJF developed the code and prepared the manuscript. GHT,
JDS and VK provided comments that shaped the final manuscript.

**Competing Interests**

The authors declare that they have no conflict of interest.

**Acknowledgements**

Daniel J. Ford was supported by a NERC GW4+ Doctoral Training Partnership studentship from the UK Natural Environment Research Council (NERC; NE/L002434/1). Gavin H. Tilstone and Vassilis Kitidis were supported by the AMT4CO₂Flux (4000125730/18/NL/FF/gp) contract from the European Space Agency and by the NERC National Capability funding to Plymouth Marine Laboratory for the Atlantic Meridional Transect (CLASS-AMT). The Atlantic Meridional Transect is funded by the UK Natural Environment Research Council through its National Capability Long-term Single Centre Science Programme, Climate Linked Atlantic Sector Science (grant number NE/R015953/1). This study contributes to the international IMBeR project and is contribution number 372 of the AMT programme. We also thank the Natural Environment Research Council Earth Observation Data Acquisition and Analysis Service (NEODAAS) for use of the Linux cluster to process the MODIS-A satellite imagery. We thank two anonymous reviewers for their comments, which have improved the manuscript.

The Surface Ocean CO₂ Atlas (SOCAT) is an international effort, endorsed by the International Ocean Carbon Coordination Project (IOCCP), the Surface Ocean Lower Atmosphere Study (SOLAS) and the Integrated Marine Biosphere Research (IMBeR) program, to deliver a uniformly quality-controlled surface ocean CO₂ database. The many researchers and funding agencies responsible for the collection of data and quality control are thanked for their contributions to SOCAT.

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
