# Peer review of "Identifying the biological control of the annual and multi-year variations in South Atlantic air-sea CO2 flux"

_Biogeosciences, 2022_

## Author Comment (AC1)

31st May 2021.

Biogeosciences.

Dear Koji Suzuki,

Thank you for your and the reviewers' comments on our manuscript entitled 'Identifying the biological control of the annual and multi-year variations in South Atlantic air-sea $CO_2$ flux' by Ford, Tilstone, Shutler and Kitidis. We have addressed all of the reviewers' comments and implemented these in the new version of the manuscript. We provide detailed responses to each of the reviewers' comments below. We hope that you find these changes satisfactory and acceptable.

We look forward to hearing from you
Yours sincerely,

[Figure]

Daniel Ford

Registered Office:
Prospect Place
The Hoe, Plymouth
PL1 3DH, UK

T   +44 (0)1752 633100
E   forinfo@pml.ac.uk
W   www.pml.ac.uk
     @PlymouthMarine

Patron: James Cameron
Registered charity number 1091222.
PML is a company limited by guarantee,
registered in England & Wales,
company number 4178503

Research excellence supporting a sustainable ocean

**Response to Anonymous Reviewer #1**

In this study, the authors assess the importance of sea surface temperature (SST) and biological activity (i.e., photosynthesis and respiration) on sea-air CO2 fluxes in the South Atlantic Ocean on seasonal and interannual scales. They used partial pressure of CO2 in the sea surface (pCO2sw) and in the atmosphere (pCO2atm), SST, NCP, NPP, estimated essentially from satellite images, as well as wind speed from reanalysis and index of climate variability modes (i.e., ENSO, SAM, NAO). Thus, they were able to correlate the differences between pCO2sw and pCO2atm and the sea-air CO2 flux with the parameters that are involved in the seasonal and interannual variation of the carbon cycle in the Atlantic Ocean. Some aspects are explored throughout the manuscript, but its main finding is that biological activity is a more important driver of interannual variation in CO2 fluxes than previously thought. This is because biological activity is generally associated with seasonal variations in CO2 flux, while large-scale physical processes are associated with long-term variations. Therefore, I consider that both the idealization of the study and its findings are relevant to the scientific community and should be encouraged for publication. However, I raised some issues that need to be improved in order to clarify some points, mainly in the discussion and in the methods used.

*Response: Thank you for your appraisal and support of our manuscript, and the comments that you have provided which have improved the manuscript. Detailed responses to each of your comments are given below.*

Overall:
Essentially, all parameters used in this study are estimated. SA-FNN uses 1o gridded SOCAT pCO2sw, which generates uncertainty in the pCO2sw estimates for every 1o gridded. pCO2atm is also estimated on a global scale. Wind speed from reanalysis is expected to underestimate in situ measurements over a wide area. Therefore, you have uncertainties in the estimates of: pCO2sw, pCO2atm, SST, salinity, wind speed. What is the uncertainty propagated by these uncertainties throughout the calculation and what is the impact on the calculated trends? The authors often warn that they considered uncertainties throughout the calculations, but it is not clear which uncertainties of each parameter were introduced in this analysis. Furthermore, nothing is mentioned about numerical uncertainties. For example, what is the average uncertainty (maximum-minimum) of the calculated trends and CO2 fluxes? My suggestion is to include a section on uncertainties and limitations and a table with uncertainties for all parameters (those that are available).

*Response: This is a good point and thank you for the suggestions on the uncertainties. In the updated manuscript, we have now included a table of the uncertainty values used for each of the parameters which are propagated through the calculations of $\Delta pCO_2$ and the $CO_2$ flux; this table is also given below for convenience:*

Table 1: Uncertainties in the input parameters used in the Monte Carlo uncertainty propagation.

| Parameter | Uncertainty | Reference |
|---|---|---|
| $pCO_{2\ (sw)}$ | Variable (Fig. B1) | (Ford et al., 2022) |
| SST | 0.441 °C | (Ford et al., 2021a) |
| SSS | 0.1 psu | (Jean-Michel et al., 2021) |
| $pCO_{2\ (atm)}$ | 1 µatm | (Takahashi et al., 2009) |
| Gas transfer velocity | 20 % | (Woolf et al., 2019) |

Registered Office:
Prospect Place
The Hoe, Plymouth
PL1 3DH, UK

T  +44 (0)1752 633100
E  forinfo@pml.ac.uk
W  www.pml.ac.uk
@PlymouthMarine

Patron: James Cameron
Registered charity number 1091222.
PML is a company limited by guarantee,
registered in England & Wales,
company number 4178503

[Figure]

Research excellence supporting a sustainable ocean

*Following your useful comments, we have now also propagated the pCO$_{2\,(atm)}$ uncertainty within the ΔpCO$_2$ analysis, and SST, SSS and pCO$_{2\,(atm)}$ uncertainties within the CO$_2$ flux analysis. The methods section 2.4 has been updated to reflect this change, and all figures reproduced to reflect the new uncertainties. The low uncertainty associated with the pCO$_{2\,(atm)}$ calculation has a small impact on the overall combined uncertainty.*

*We have added an Appendix B to the manuscript to show the uncertainties in the SA-FNN pCO$_{2\,(sw)}$ and the associated trends as two new figures. The first figure (Figure B1) presents the mean and standard deviation of the SA-FNN pCO$_{2\,(sw)}$ uncertainty, which highlights regions where the uncertainties are higher (a) and more variable (b). This figure (Figure B1) is given here:*

[Figure]

Figure B1 – (a) Mean SA-FNN pCO$_{2\,(sw)}$ uncertainty between July 2002 and December 2018. Longhurst provinces (Longhurst, 1998) used within the SA-FNN training described in Ford et al. (2022). The province areas acronyms are listed as follows:western tropical Atlantic is WTRA; eastern equatorial Atlantic is ETRA; South Atlantic Gyre is SATL; Brazilian current coastal is BRAZ; Benguela Current coastal upwelling is BENG; Southwest Atlantic shelves is FKLD; South Subtropical Convergence is SSTC; sub-Antarctic is SANT and Antarctic is ANTA. (b) Standard deviation of SA-FNN pCO$_{2\,(sw)}$ uncertainty.

*The second figure (Figure B2) addresses the uncertainty in the ΔpCO$_2$, and CO$_2$ flux trends introduced by propagating the parameter uncertainties that are given in Table 1. Instead of providing the maximum and minimum uncertainties we have mapped the uncertainties on a per pixel basis, to display where the uncertainties in the trends are higher. Figure B2 is displayed below:*

Registered Office:
Prospect Place
The Hoe, Plymouth
PL1 3DH, UK

T  +44 (0)1752 633100
E  forinfo@pml.ac.uk
W  www.pml.ac.uk
🐦 @PlymouthMarine

Patron: James Cameron
Registered charity number 1091222.
PML is a company limited by guarantee,
registered in England & Wales,
company number 4178503

[Figure]

[Figure]

Figure B2 – (a) Uncertainty in the $\Delta p CO_2$ trends presented in Fig. 5a. (b) Uncertainty in the air-sea $CO_2$ flux trends presented in Fig. 5b.

*We have added an additional section (section 2.6) to the methodology to address the limitations. The text in section 2.4 describing the limitations has been moved to section 2.6. The uncertainties propagated at each stage of the analysis are described in section 2.4 and 2.5 alongside the analysis conducted so that it is clear what uncertainties were included. The uncertainty values are presented in Table 1 and referred to within the text of sections 2.4 and 2.5.*

My second concern is about the description of methods. Despite indicating the references to the analyses carried out, the description of the methods used is superficial, making it difficult to assess them further. For example, you cite Henson et al. (2018) for a detailed description of the analysis of seasonal and interannual $\Delta p CO2$ drivers, but they do not. In fact, the method is originally described in another study (i.e., Shiskin et al., 1967) but has been adapted for $\Delta p CO2$. Here it is essential that the assumptions and adaptations, as well as the limitations, are described, since this is the main analysis of the study from which the discussion is carried out.

*Response: This is a good point and we have now provided a more thorough description of the X-11 analysis which describes the three main steps to determining the seasonal and interannual components used in the analysis. We refer the reader to Shiskin et al (1967) and Pezzuli et al. (2005) where the X-11 method applied to environmental timeseries is first described, and the modifications there in. The text now reads "The X-11 analytical econometric tool (Shiskin et al., 1967) was used to decompose the timeseries into seasonal, interannual and residual components following the methodology of Pezzulli et al. (2005). In brief, the X-11 method comprises a three step filtering algorithm; (1) The interannual component ($T_t$) is initially estimated using an annual centred running mean, which is subtracted from the initial timeseries ($X_t$) to estimate the seasonal component ($S_t$). (2) $T_t$ is revised by applying an annual centred running mean to the $X_t$ minus $S_t$. The revised $T_t$ is removed from $X_t$ and the final $S_t$ calculated. (3) The final $T_t$ is calculated by applying an annual centred*

Registered Office:
Prospect Place
The Hoe, Plymouth
PL1 3DH, UK

T  +44 (0)1752 633100
E  forinfo@pml.ac.uk
W  www.pml.ac.uk
   @PlymouthMarine

Patron: James Cameron
Registered charity number 1091222.
PML is a company limited by guarantee,
registered in England & Wales,
company number 4178503

*running mean to $X_t$ minus the revised $S_t$. The analysis has been shown to be effective in the decomposition of environmental time-series (Pezzulli et al., 2005; Vantrepotte & Mélin, 2011; Henson et al., 2018), that allows the seasonal cycle to vary on a yearly basis and, produces an interannual component that results in a robust representation of the longer-term changes in the timeseries."*
*We have now removed the reference to Henson et al. (2018) as being the sole methodology used, as we realised (thanks to the reviewer's query) that this really only describes the methodology that was followed in Appendix A for constructing the in situ $\Delta pCO_2$ timeseries and so this section does still reference Henson et al., (2018).*

I missed the influence of salinity, especially in regions where freshwater input by rivers and rain are significant, for example western Tropical Atlantic and southwestern South Atlantic. Although biological activity significantly influences pCO2sw, the dilution of seawater by riverine water directly influences pCO2sw, via mixing and solubility. This to some extent must be being counted as "biological activity" here. It might be interesting to use salinity as a parameter for correlation with $\Delta pCO2$ and CO2 flux instead of NAO, which was neither significant nor discussed here. Or even if they do not make the correlation with salinity, I think it will be important to include it in the discussion of these commented regions.

***Response:*** *Thank you for the suggestions which we agree with. We have removed correlations to the NAO from the manuscript, as you highlight, they are neither significant nor discussed. We have replaced the NAO data with sea surface salinity (SSS) data within the X-11 analysis as suggested. We have now included descriptions and discussion on the key correlations between SSS, $\Delta pCO_2$ and $CO_2$ flux within the results and discussion (within sections 3.1, 3.2, 4.1 and 4.2). The full detailed text is also given below within the responses to the reviewer's specific comments. See below the updated figures 1-4 that now include SSS.*

[Figure]

Figure 1: Significant Spearman correlations between the $\Delta pCO_2$ seasonal component of the X-11 analysis and (a) net community production (NCP), (b) net primary production (NPP), (c) sea surface temperature (SST), (d) sea surface salinity (SSS), (e) Multivariate ENSO index (MEI) and (f) Southern Annular Mode (SAM) seasonal

Registered Office:
Prospect Place
The Hoe, Plymouth
PL1 3DH, UK

T +44 (0)1752 633100
E forinfo@pml.ac.uk
W www.pml.ac.uk
@PlymouthMarine

Patron: James Cameron
Registered charity number 1091222.
PML is a company limited by guarantee,
registered in England & Wales,
company number 4178503

components. White regions indicate no significant correlations, and green regions indicate no analysis was performed due to missing satellite data.

[Figure]

Figure 2: Significant Spearman correlations between the air-sea $CO_2$ flux seasonal component of the X-11 analysis and (a) net community production (NCP), (b) net primary production (NPP), (c) sea surface temperature (SST), (d) sea surface salinity (SSS), (e) Multivariate ENSO index (MEI) and (f) Southern Annular Mode (SAM) seasonal components. White regions indicate no significant correlations, and green regions indicate no analysis was performed due to missing satellite data.

[Figure]

Figure 3: Significant Spearman correlations between the $\Delta pCO_2$ interannual component of the X-11 analysis and (a) net community production (NCP), (b) net primary production (NPP), (c) sea surface temperature (SST), (d) sea surface salinity (SSS), (e) Multivariate ENSO index (MEI) and (f) Southern Annular Mode (SAM)

Registered Office:
Prospect Place
The Hoe, Plymouth
PL1 3DH, UK

T  +44 (0)1752 633100
E  forinfo@pml.ac.uk
W  www.pml.ac.uk
   @PlymouthMarine

Patron: James Cameron
Registered charity number 1091222.
PML is a company limited by guarantee,
registered in England & Wales,
company number 4178503

interannual components. White regions indicate no significant correlations, and green regions indicate no analysis was performed due to missing satellite data.

[Figure]

Figure 4: Significant Spearman correlations between the air-sea $CO_2$ flux interannual component of the X-11 analysis and (a) net community production (NCP), (b) net primary production (NPP), (c) sea surface temperature (SST), (d) sea surface salinity (SSS), (e) Multivariate ENSO index (MEI) and (f) Southern Annular Mode (SAM) interannual components. White regions indicate no significant correlations, and green regions indicate no analysis was performed due to missing satellite data.

Specific comments:
Title: Please consider changing the title to include "seasonal and interannual" variations instead of "interannual and long-term".

*Response: Thank you for the suggestion. We have modified the title to "Identifying the biological control of the annual and multi-year variations in South Atlantic air-sea $CO_2$ flux"*

Introduction:

The Introduction is very well written and addresses the main problem that was investigated.

- Line 30: Here are you sure you mean 'sequestering' (i.e., stored in the deep ocean) or CO2 uptake by the sea surface? It might be interesting to indicate the percentage of how much this value represents, as you do not mention anything else about it throughout the introduction, so this value alone does not make clear the real importance of the oceans in sequestering atmospheric CO2.

*Response: Thank you for the clarification. Yes, we are referring to the $CO_2$ uptake at the sea surface, and therefore have corrected the sentence to "The global oceans have buffered the rise by acting as a sink for atmospheric $CO_2$ at a rate of between 1 and 3.5 Pg C yr$^{-1}$ (e.g. Friedlingstein et al., 2020; Landschützer et al., 2014; Watson et al., 2020)."*

[Figure]

Registered Office:
Prospect Place
The Hoe, Plymouth
PL1 3DH, UK

T  +44 (0)1752 633100
E  forinfo@pml.ac.uk
W  www.pml.ac.uk
   @PlymouthMarine

Patron: James Cameron
Registered charity number 1091222.
PML is a company limited by guarantee,
registered in England & Wales,
company number 4178503

Research excellence supporting a sustainable ocean

- Line 38: The solubility of CO2 and the CO2 flux are also directly influenced by the sea surface salinity.

**Response:** *Apologies, there was a mistyping in our initial sentence because $\Delta pCO_2$ does not contain a solubility term. This sentence has now been corrected and reads: "$\Delta pCO_2$ can therefore be controlled by changes in sea surface temperature (SST), because the $pCO_2$ is proportional to the temperature.". The solubility of $CO_2$ is temperature and salinity dependent and we have corrected this in a later comment on the air-sea $CO_2$ flux.*

- Line 50: The solubility coefficient used in calculating the CO2 flux is also a function of both temperature and salinity.

**Response:** *We have corrected the sentence to include the vertical haline gradients which are discussed in Woolf et al. (2016) and implemented in this study. This sentence now reads: "The $CO_2$ concentration difference is determined by the $pCO_2$ at the base ($pCO_{2 (sw)}$) and top ($pCO_{2 (atm)}$) of the mass boundary layer and the respective solubilities (Weiss, 1974), which must be carefully calculated due to vertical thermo-haline gradients existing across the mass boundary layer (Woolf et al., 2016)."*

- Line 57: non-SST instead of non-temperature.
- Line 60: SST and non-SST instead of temperature and non-temperature.

**Response:** *We have changed the wording as suggested.*

Methods
- Line 76: Ford et al. (2021b) is cited here, but there is no earlier citation for Ford et al., 2021. Typo?

**Response:** *Apologies, we have now corrected this typo throughout the manuscript.*

-Line 80: "into eight static provinces in the South Atlantic Ocean". Which provinces are these and why are they important for the development of AS-FNN? By analysing each province separately, the AS-FNN can better capture regional variations than if the region were analysed as a whole, is that it? If so, are there significant differences in uncertainties between different provinces that could impact the interpretation of results for some of them?

**Response:** *Yes, this is correct, the use of 8 static provinces allows the SA-FNN to reproduce the regional $pCO_{2 (sw)}$ variability in the South Atlantic Ocean. The inclusion of provinces either static or variable is common practice within methods that extrapolate global of $pCO_{2 (sw)}$ (e.g. Landschützer et al., 2014; Denvil-Sommer et al., 2019). The provinces referred to here are the Longhurst provinces (Longhurst, 1998) in the South Atlantic Ocean, with modifications as discussed in Ford et al. (2022; i.e. merging of the WTRA and ETRA into a single province). Longhurst provinces are widely accepted oceanographic domains. We have included a map of the Longhurst provinces within a new appendix figure (Figure B1; displayed below) that also displays the $pCO_{2 (sw)}$ uncertainties.*
*The uncertainties in $pCO_{2 (sw)}$ estimated from the SA-FNN are spatially and temporally varying, therefore Figure B1 displays the mean and standard deviation SA-FNN uncertainty for each 1 deg pixel. There are regions of higher uncertainty such as the Amazon Plume and north Brazil coast, but*

Registered Office:
Prospect Place
The Hoe, Plymouth
PL1 3DH, UK

T  +44 (0)1752 633100
E  forinfo@pml.ac.uk
W  www.pml.ac.uk
🐦 @PlymouthMarine

Patron: James Cameron
Registered charity number 1091222.
PML is a company limited by guarantee,
registered in England & Wales,
company number 4178503

*uncertainties remain relatively consistent between different provinces. This ammendment also addresses an earlier reviewer comment.*
*This figure (Figure B1) is also displayed below:*

[Figure]

Figure B1 – (a) Mean SA-FNN $pCO_{2\ (sw)}$ uncertainty between July 2002 and December 2018. Longhurst provinces (Longhurst, 1998) used within the SA-FNN training described in Ford et al. (2022; note the WTRA and ETRA are merged into one province). The province areas acronyms are listed as follows: WTRA is western tropical Atlantic; ETRA is eastern equatorial Atlantic; SATL is South Atlantic Gyre; BRAZ is Brazilian current coastal; BENG is Benguela Current coastal upwelling; FKLD is Southwest Atlantic shelves; SSTC is South Subtropical Convergence; SANT is sub-Antarctic and ANTA is Antarctic. (b) Standard deviation of SA-FNN $pCO_{2\ (sw)}$ uncertainty.

- Line 81: How and what is the impact of pCO2atm on pCO2sw in these estimates?

***Response:*** *The use of $pCO_{2\ (atm)}$ within the SA-FNN allows the machine learning approach to include a signal that provides the rising atmospheric $CO_2$ without the need to fix a rate of increase (which would bias the SA-FNN estimates). $pCO_{2\ (atm)}$ is spatially homogeneous compared to $pCO_{2\ (sw)}$, so the effect on the spatial $pCO_{2\ (sw)}$ component is small. This is common practice within methods to globally extrapolate $pCO_{2\ (sw)}$ such as Landschutzer et al. (2014) and Watson et al. (2020).*

- Line 85: Here you mean that the pCO2atm used was extracted from the product of Landschützer et al. (2016, 2017) and it was in turn estimated from NOAA-ESRL regional stations? This is not clear to me. Perhaps "Monthly 1o grids of pCO2(atm) were extracted from v5.5 of the global estimates of pCO2(sw) dataset (Landschützer et al., 2016, 2017), which was estimated using the dry mixing ratio of CO2 from the NOAA -ESRL marine boundary layer reference (https://www.esrl.noaa.gov/gmd/ccgg/mbl/; last accessed 25/09/2020), Optimum Interpolated SST (Reynolds et al., 2002) and sea level pressure following Dickson et al. (2007)."

***Response:*** *This is correct, and we have modified the wording as suggested.*

- Line 94: Why did you use Nightingale et al. (2000) and not that of Ho et al. (2006) or Wanninkhof (2014), which have been shown to be more appropriate for ocean CO2 flux calculations? If there is no reasonable explanation for using this parameterization, I strongly recommend using Ho et al. (2006) or Wanninkhof (2014), as they are more appropriate for oceanic regions.

*Response: The parameterizations of Ho et al. (2006) and Wanninkhof (2014) are similar to that of Nightingale et al. (2000). The three parameterization are plotted in the figure below, with a ± 20% uncertainty as described in Woolf et al. (2019) and Wanninkhof (2014). All three parameterizations are within the uncertainties of the Wanninkhof (2014) parameterization, and follow a similar profile. We have therefore decided to use Nightingale et al. (2000), and hope you understand the reasoning for this.*

[Figure]

- Line 99: This information has already been described above in line 85.

*Response: The information on this line is slightly different to the pCO$_{2 (atm)}$ retrieved from the Landschützer et al. (2017) product. This description accounts for the vertical temperature gradients that affect the CO$_2$ solubility for the air-sea CO$_2$ flux calculations, and the recalculation of pCO$_{2 (atm)}$ using skin SST. This text is important for the methodology, and therefore we have kept this text. We hope you understand the reasoning for this.*

- Line 103: I was wondering why you set the time series from 2002 to 2018. Is it because of limited data availability, I suppose?

*Response: This is correct, the SA-FNN pCO$_{2 (sw)}$ dataset retrieved from Ford et al. (2021a) is for the time period July 2002 to December 2018.*

[Figure]

Registered Office:
Prospect Place
The Hoe, Plymouth
PL1 3DH, UK

T  +44 (0)1752 633100
E  forinfo@pml.ac.uk
W  www.pml.ac.uk
  @PlymouthMarine

Patron: James Cameron
Registered charity number 1091222.
PML is a company limited by guarantee,
registered in England & Wales,
company number 4178503

- Line 106: Chlorophyll-a (Chl a).

*Response: We have corrected the definition of Chl a.*

- Line 121: A more detailed description of the analysis of seasonal and interannual ΔpCO2 drivers is needed. Please see general comment.

*Response: We have updated the text in section 2.4 to include a description of the basic steps the X-11 analysis conducts to retrieve the seasonal and interannual components. We have also corrected the references as per your general comment. This updated text reads "The X-11 analytical econometric tool (Shiskin et al., 1967) was used to decompose the timeseries into seasonal, interannual and residual components following the methodology of Pezzulli et al. (2005). In brief, the X-11 method comprises a three step filtering algorithm; (1) The interannual component ($T_t$) is initially estimated using an annual centred running mean, which is subtracted from the initial timeseries ($X_t$) to estimate the seasonal component ($S_t$). (2) $T_t$ is revised by applying an annual centred running mean to the $X_t$ minus $S_t$. The revised $T_t$ is removed from $X_t$ and the final $S_t$ calculated. (3) The final $T_t$ is calculated by applying an annual centred running mean to $X_t$ minus the revised $S_t$. The analysis has been shown to be effective in the decomposition of environmental time-series (Pezzulli et al., 2005; Vantrepotte & Mélin, 2011; Henson et al., 2018), that allows the seasonal cycle to vary on a yearly basis and, produces an interannual component that results in a robust representation of the longer-term changes in the timeseries."*

- Line 124: Despite indicating that an error propagation analysis from pCO2sw was performed, no results regarding this are shown. Also, if I understood correctly, you just computed the uncertainty of the pCO2sw estimate in the ΔpCO2 and CO2 flux calculation, right? If so, it is important to account for the uncertainties in all other estimated parameters that go into the CO2 flux calculation (i.e., SST, salinity, wind speed, pCO2atm), especially wind speed. Although reanalysis wind speed data is commonly used as a proxy for in situ data in comparisons with model outputs, the wind speed of reanalyses underestimates the in situ wind speed. I do not think this propagated error assessment is too problematic. However, if this seems like too much work, perhaps a sensitivity analysis is appropriate. For example, by calculating the CO2 flux with the highest expected uncertainty for all parameters, then you will have an overestimate of the propagation of uncertainties throughout the CO2 flux calculation.

*Response: The spatially and temporally variable uncertainties of the SA-FNN $pCO_{2\,(sw)}$ were determined within Ford et al. (2022), and these uncertainties are contained within the SA-FNN dataset (Ford et al. 2021a). We referred the reader to Ford et al. (2022) for a full description of how these uncertainties are determined.*
*In this study, we propagated the SA-FNN $pCO_{2\,(sw)}$ uncertainty to $\Delta pCO_2$, and then through the X-11 and drivers analysis using a Monte Carlo uncertainty propagation as described in section 2.4. For the $CO_2$ flux analysis, we propagate the SA-FNN $pCO_{2\,(sw)}$ and the gas transfer velocity uncertainties through the $CO_2$ flux calculations, and then through the X-11 and drivers' analysis.*
*In response to the reviewer comments, we have now also propagated the $pCO_{2\,(atm)}$ uncertainty within the $\Delta pCO_2$ analysis. This had no observable effect on the results, due to the $pCO_{2\,(sw)}$ estimates being much more spatially and temporally variable than $pCO_{2\,(atm)}$. For the $CO_2$ flux analysis, we have now*

[Figure]

Registered Office:
Prospect Place
The Hoe, Plymouth
PL1 3DH, UK

T +44 (0)1752 633100
E forinfo@pml.ac.uk
W www.pml.ac.uk
@PlymouthMarine

Patron: James Cameron
Registered charity number 1091222.
PML is a company limited by guarantee,
registered in England & Wales,
company number 4178503

*also propagated the uncertainties in SST, SSS and $pCO_{2\ (atm)}$. Similar to the $\Delta pCO_2$ analysis, including these uncertainties had no observable effect on the results, as the $pCO_{2\ (sw)}$ and gas transfer velocity uncertainties dominate the analysis (a finding that is consistent with the work of Woolf et al., 2019). To clearly identify the uncertainties in each parameter we now have included a table within section 2.4, that states the uncertainty and the relevant reference. This table was added as it addresses a previous reviewer request and it is also given below:*

Table 1: Uncertainties in the input parameters used in the Monte Carlo uncertainty propagation.

| Parameter | Uncertainty | Reference |
|---|---|---|
| $pCO_{2\ (sw)}$ | Variable (Fig. B1) | (Ford et al., 2022) |
| SST | 0.441 °C | (Ford et al., 2021a) |
| SSS | 0.1 psu | (Jean-Michel et al., 2021) |
| $pCO_{2\ (atm)}$ | 1 µatm | (Takahashi et al., 2009) |
| Gas transfer velocity | 20 % | (Woolf et al., 2019) |

- Line 130: You considered the North Atlantic Oscillation (NAO), but did not mention the reason for it, as you did with ENSO and SAM. How can NAO influence pCO2sw and CO2 flux in the South Atlantic? What were you expected to find and what did you find in relation to this mode of climate variability?

***Response:*** *As suggested in a previous reviewer comment, we have now removed the analysis of the NAO replacing it with SSS. For interest, the NAO was initially included due to the correlation between the NAO and NCP identified by Tilstone et al. (2015), and could suggest a teleconnection between the North and South Atlantic Oceans.*

- Line 137: It is important to show at some point the values of propagated uncertainties. 10% was the gas transfer coefficient uncertainty (i.e., wind speed) propagated in the CO2 flux calculation? A 10% certainty for this coefficient seems very low, especially when it represents 70% of the uncertainty of the CO2 flux and using reanalysis data. I suggest making a table, which can be for the supplementary material, with the uncertainties of each of the parameters used (i.e., SST, salinity, pCO2sw, pCO2atm, wind speed, NPP, NCP).

***Response:*** *Please see previous comments on this point. We have now included a table of uncertainties for each of parameters used within the analysis in the manuscript, which is given below. As you highlight, this is a key piece of information to include for the analysis. We have increased the uncertainty in the gas transfer velocity to ±20%, as suggested by Woolf et al. (2019), in response to your comments.*

Table 1: Uncertainties in the input parameters used in the Monte Carlo uncertainty propagation.

| Parameter | Uncertainty | Reference |
|---|---|---|
| $pCO_{2\ (sw)}$ | Variable (Fig. B1) | (Ford et al., 2022) |
| SST | 0.441 °C | (Ford et al., 2021a) |
| SSS | 0.1 psu | (Jean-Michel et al., 2021) |
| $pCO_{2\ (atm)}$ | 1 µatm | (Takahashi et al., 2009) |
| Gas transfer velocity | 20 % | (Woolf et al., 2019) |

Registered Office:
Prospect Place
The Hoe, Plymouth
PL1 3DH, UK

T +44 (0)1752 633100
E forinfo@pml.ac.uk
W www.pml.ac.uk
@PlymouthMarine

Patron: James Cameron
Registered charity number 1091222.
PML is a company limited by guarantee,
registered in England & Wales,
company number 4178503

- Line 138: Could this be resolved by doing the analysis with the pCO2sw normalised by the SST? So, you should find a higher correlation between ΔpCO2 and NCP while the correlation between ΔpCO2 and SST would decrease.

*Response: Thank you for the suggestion. By normalizing the $pCO_{2\ (sw)}$ to the mean SST the correlations to the drivers would reverse in sign, but their magnitudes would stay relatively similar.*

- Line 145: That seems appropriate. If the correlation decreases using in situ pCO2sw, can this indicate how much the estimated ΔpCO2 is biased by the SST?

*Response: The aim of this analysis was to confirm that the SA-FNN retrieved drivers were consistent with the in situ observations, acknowledging the limitations of the spatial and temporal averaging as described in Appendix A. Applying the approach to in situ $pCO_{2\ (sw)}$ would however not indicate whether the $ΔpCO_2$ is biased by the SST due to the spatial and temporal averaging of the in situ $pCO_{2\ (sw)}$ timeseries. For example, missing values in the in situ $pCO_{2\ (sw)}$ timeseries are filled with mean values from the same month, where as the satellite SST timeseries is complete. Therefore if SST were higher in a particular month and year, where the mean $pCO_{2\ (sw)}$ were used then the correlation will likely decrease. To evaluate whether a bias existed this would require a complete in situ $pCO_{2\ (sw)}$ timeseries.*

Results
- Line 161: Was the correlation between ΔpCO2 and NCP in the equatorial region numerically greater or was the area of significant correlation greater? If you only consider the area with significant correlation, is the correlation between ΔpCO2 and NCP higher? This could be made clearer if you showed the number values of the correlations in each region in the text.

*Response: We have now removed this sentence, as we can see it was confusing and was not discussed further.*

- Line 165: In the northern part of the Brazil Current (~12ºS-17ºS) there is a more intense positive correlation between ΔpCO2 and NPP, in contrast to the surrounding waters where the correlation is negative. Is there any suggested explanation for this?

*Response: Yes and we have clarified this in the text. This correlation is likely to be due to upwelling as identified by Aguiar et al. (2018) in this region. We have added a sentence in the results and a paragraph in the discussion which reads "At between 12° S and 17 °S along the South American coast, there were also deviations from the expected drivers as there were positive correlations between NPP and $ΔpCO_2$ (Fig. 1b) and negative correlations between SSS and $ΔpCO_2$ (Fig. 1d), which are consistent with an upwelling signature that occurs along the coast. Aguiar et al. (2018) also showed intense seasonal upwelling events in this region that are driven by wind and currents."*

- Line 170. There is a band with a positive correlation sign south of 40ºS for NCP and NPP. (Fig. 1 a, b). This is also true of the southern coast of South America. This appears to be as important as the regions under the influence of the Amazon River plume and the Benguela upwelling. However, none of this is mentioned here or discussed later.

Registered Office:
Prospect Place
The Hoe, Plymouth
PL1 3DH, UK

T +44 (0)1752 633100
E forinfo@pml.ac.uk
W www.pml.ac.uk
🐦 @PlymouthMarine

Patron: James Cameron
Registered charity number 1091222.
PML is a company limited by guarantee,
registered in England & Wales,
company number 4178503

*Response: Thank you for highlighting this.*

I suggest as a point of discussion:

"Between 30°-45°S, dissolved inorganic carbon and SST exert a similar influence on pCO2sw, indicating that seasonal changes in dissolved inorganic carbon driven by biological uptake in the summer and upwelling in winter are approximately balanced by seasonal changes in SST and their control on the solubility pump." (Henley et al., 2020).

The southern coast of South America is strongly influenced by riverine water input that dilutes the total alkalinity when it mixes with seawater, leading to an increase in pCO2sw (Liutti et al., 2020). This is associated with a supply of nutrients, which increases photosynthesis, however the main drivers of pCO2sw in this region are total alkalinity and SST (Liutti et al., 2020). This likely explains the positive correlation between ΔpCO2 and both NCP and NPP.

*Response: Thank you for the suggested discussion points which we agree with. We have now clearly identified these regions in the results section (3.1) which reads "The South American coast between 12 °S and 17 °S displayed positive correlations between $\Delta pCO_2$ and NPP (Fig. 1b), along with negative correlations between $\Delta pCO_2$ and SSS (Fig. 1e). Negative correlation between $\Delta pCO_2$ and SSS, and positive correlations between NCP, NPP and $\Delta pCO_2$ were also observed in the southwestern Atlantic (Fig. 1e). Positive correlations between NCP, NPP and $\Delta pCO_2$ were identified in a band across 40 °S (Fig. 1a, b)."*
*In the discussion section 4.1 we have now included the following text, which is a modified version of the suggestions provided. This text reads "Between 30 °S and 45 °S, dissolved inorganic carbon and SST exert a similar influence on $pCO_{2\,(sw)}$, indicating that seasonal changes in dissolved inorganic carbon driven by biological uptake in the summer and upwelling in winter are approximately balanced by seasonal changes in SST and their control on the solubility pump (Henley et al., 2020). This likely explains the band of positive correlations between NCP, NPP and $\Delta pCO_2$ and sharp transitions in correlations between SST and $\Delta pCO_2$ across ~40 °S."*
*and*
*"At between 12° S and 17 °S along the South American coast, there were also deviations from the expected drivers as there were positive correlations between NPP and $\Delta pCO_2$ (Fig. 1b) and negative correlations between SSS and $\Delta pCO_2$ (Fig. 1d), which are consistent with an upwelling signature that occurs along the coast. Aguiar et al. (2018) also showed intense seasonal upwelling events in this region that are driven by wind and currents. The southern coast of South America is strongly influenced by riverine water input that reduces the total alkalinity and therefore causes an increase in $pCO_{2\,(sw)}$ (Liutti et al., 2021). This is associated with an increased supply of nutrients which in turn enhances NPP, though the main drivers of $pCO_{2\,(sw)}$ in this region still remain as total alkalinity and SST (Liutti et al., 2021). This potentially explains the positive correlation between $\Delta pCO_2$ and both NCP and NPP (Fig. 1a, b), as well as the negative correlations between $\Delta pCO_2$ and SSS. The extension offshore of this negative correlation between SSS and $\Delta pCO_2$ (Fig. 1d) could be caused by the advection of water masses due to intense mesoscale eddy activity arising from the Brazil-Malvinas confluence (Mason et al., 2017)."*

- Line 200: Since NCP responds to processes that occur essentially in the ocean, how do you explain the correlation being greater with CO2 flux than with ΔpCO2? This indicates that this correlation is

essentially associated with interannual variability in wind speed, correct? It seems that the correlation you are finding here is between NCP/NPP and wind speed and not with CO2 flux (which implies influence on pCO2sw). Perhaps somehow the NPC/NPP estimates are biased by wind speed?

***Response:*** *We have clarified this within the text. Ford et al. (2021b) showed that wind speeds anomalies were positively correlated with NCP anomalies in the South Atlantic gyre. This indicates that the correlations between $CO_2$ flux and NCP are enhanced by wind speed since wind speed is integral to the calculation of $CO_2$ flux. It should be noted that $CO_2$ flux is a linear function of the air-sea $CO_2$ concentration gradient, but a quadratic function of wind speed. A doubling in wind speed may thereby drive a larger $CO_2$ flux than a doubling in the air-sea $CO_2$ gradient as shown previously for the northwest European shelf (Kitidis et al., 2019). This relationship also highlights that an increase in NCP, which causes a biological drawdown of $CO_2$, contributes to an enhancement of the difference in $CO_2$ concentration. This increased difference is acted upon by the increased gas transfer at higher wind speeds. This also applies in reverse, and therefore shows an important mechanism for biological amplification of changes in the $CO_2$ flux. This is discussed in the context of changes in NCP and wind speeds driven by the MEI which reads "Positive correlations between the MEI and $CO_2$ flux (Fig. 4d) indicate that the MEI partially controls the interannual variability in $CO_2$ flux in the South Atlantic subtropical gyre, through modulations primarily in SST and to a lesser extent NCP. The South Atlantic Subtropical Anticyclone has been observed to strengthen (weaken) and move south (north) during La Niña (El Niño) events. This displacement increases (decreases) wind speeds across the subtropical South Atlantic, which will enhance (weaken) gas exchange, and elevate (depress) NCP (Ford et al., 2021b). These results suggest a more significant role of NCP in controlling the interannual variability in the $CO_2$ flux than previously thought."*

Where are the correlations between ΔpCO2 and both NCP and NPP significant indicating that much of the carbon produced at the surface is being exported to the deep ocean? Conversely, where the correlation between ΔpCO2 and NCP is higher (line 161) does that mean that surface production is being advected to another region or is NCP not produced locally? That makes sense?

***Response:*** *Regions where either NCP, NPP or both are significantly correlated to $\Delta pCO_2$ indicates a potential biological contribution to the $\Delta pCO_2$ variability. In regions where NCP has a higher magnitude correlation to $\Delta pCO_2$ compared to NPP would suggest that respiration variability is increasing the correlations to $\Delta pCO_2$, for example in the South Atlantic gyre where Serret et al. (2015) showed that both NPP and respiration control NCP variability. From our data we cannot assess the carbon that is exported to depth.*

- Line 236: SST instead of temperature.

***Response:*** *We have changed the wording as suggested.*

- Line 239: Only the correlation between these parameters (NCP, NPP, SST) and ΔpCO2 and CO2 flux do not necessarily indicate the greater influence of biology or temperature. In Fig. 1 SST is well correlated with ΔpCO2 and CO2 flux over virtually the entire region and only the sign of the correlation changes. For example, if the correlation is -0.6 or 0.6 the intensity of the correlation is the same, only the sign changes. So, the statement that "biological activity was a key driver of seasonal variability in response to the equatorial upwelling and highlighting the dominance of non-temperature

[Figure]

Registered Office:
Prospect Place
The Hoe, Plymouth
PL1 3DH, UK

T  +44 (0)1752 633100
E  forinfo@pml.ac.uk
W  www.pml.ac.uk
    @PlymouthMarine

Patron: James Cameron
Registered charity number 1091222.
PML is a company limited by guarantee,
registered in England & Wales,
company number 4178503

Research excellence supporting a sustainable ocean

drivers" does not seem to me to be supported by the correlations in Fig. 1. This would be more evident if the average coefficient of determination (R2) value were shown.

*Response: We have now removed the end of this sentence. As you have mentioned the intensity of the correlation does not necessarily indicate the dominance of either temperature or NCP. But the correlations do support the conclusion that NCP is likely an important component of the seasonal variability in the eastern Equatorial Atlantic in response to the equatorial upwelling. This sentence now reads "We found positive correlations between the NCP, ΔpCO₂ and CO₂ flux seasonal components, indicating that biological activity is likely a key driver of seasonal variability in response to the equatorial upwelling."*

- Line 242: Instead of "biological activity" I suggest indicating the specific process you are referring to (e.g., photosynthesis, respiration) because this should change according to the sign of the correlation between pCO2sw and both NCP and NPP and "biological activity" is a broad term that suggests both a decrease in and an increase in CO2.
- Line 294: Again, be more specific about which biological activity you are referring to, photosynthesis or respiration.

*Response: Thank you for the suggestion, we have clarified these. The analysis correlates changes in NCP/NPP to changes in ΔpCO₂ and the CO₂ flux, determining the driving component between photosynthesis or respiration is not possible. For example, in the South Atlantic gyre, where negative correlations between NCP/NPP and ΔpCO₂ are observed on interannual timescales, an increase in NCP/NPP is associated with a decrease in ΔpCO₂. But conversely a decrease in NCP/NPP is associated with an increase in ΔpCO₂. We have therefore changed "biological activity" to "NCP", as NCP quantifies the balance between photosynthesis and respiration, and is more precise.*

- Line 244: 0.76 instead of -0.76 for o R2.

- Line 245: 0.13 instead of -0.13 for o R2.

*Response: We have corrected these two statements to R values as reported by Lefèvre et al. (2016).*

- Line 271: This information should be repositioned to the Material and Methods section with more detailed information about the method. For example, if possible, what calculation is done to extract seasonal and interannual cycles, what adaptations were made from the original econometric analysis, and what are its limitations.

*Response: As a result of an earlier reviewer request, the paragraph describing the benefits of using the X-11 analysis has been moved to the section 2.4, and a brief description of the calculations required to extract the seasonal and interannual components. This reads: "The X-11 analytical econometric tool (Shiskin et al., 1967) was used to decompose the timeseries into seasonal, interannual and residual components following the methodology of Pezzulli et al. (2005). In brief, the X-11 method comprises a three step filtering algorithm; (1) The interannual component ($T_t$) is initially estimated using an annual centred running mean, which is subtracted from the initial timeseries ($X_t$) to estimate the seasonal component ($S_t$). (2) $T_t$ is revised by applying an annual centred running mean to the $X_t$ minus $S_t$. The revised $T_t$ is removed from $X_t$ and the final $S_t$ calculated. (3) The final $T_t$ is*

Registered Office:
Prospect Place
The Hoe, Plymouth
PL1 3DH, UK

T  +44 (0)1752 633100
E  forinfo@pml.ac.uk
W  www.pml.ac.uk
    @PlymouthMarine

Patron: James Cameron
Registered charity number 1091222.
PML is a company limited by guarantee,
registered in England & Wales,
company number 4178503

Research excellence supporting a sustainable ocean

*calculated by applying an annual centred running mean to $X_t$ minus the revised $S_t$. The analysis has been shown to be effective in the decomposition of environmental time-series (Pezzulli et al., 2005; Vantrepotte & Mélin, 2011; Henson et al., 2018), that allows the seasonal cycle to vary on a yearly basis and, produces an interannual component that results in a robust representation of the longer-term changes in the timeseries."*

- Line 284: Use verbs in the present tense to refer to the findings (conclusions) of the studies. For example: "seasonal and interannual drivers of ΔpCO2 are different" instead of "were". On the other hand, use verbs in the past tense to refer to results.

***Response:*** *We have corrected this to are. The sentence now reads "In the North Atlantic Ocean, Henson et al. (2018) showed that the seasonal and interannual drivers of $\Delta pCO_2$ are different, which could arise from the necessity to study $CO_2$ fluxes over longer timescales."*

Line 296: Something does not seem to make sense here. A negative correlation between MEI and CO2 flux implies that CO2 exchange is more intense during La Niña (when the ENSO index is negative), correct?
- Line 305: Same as the previous comment here. With negative SAM and the migration of westerly winds further north, should not the CO2 flux be increasing rather than decreasing?

***Response:*** *We have clarified these sentences within the text. The $CO_2$ flux can be either positive ($CO_2$ source) or negative ($CO_2$ sink), and therefore a negative correlation between the MEI and $CO_2$ flux would show that the $CO_2$ flux is moving towards a source (i.e becoming a weaker sink) during La Niña phases. We have clarified this in the text which reads "The negative correlation between the $CO_2$ flux and the MEI in a band between 30° S and 45° S (Fig. 4e), indicates that reduced (elevated) wind speeds that occur during La Niña (El Niño) events in this region, suppress (enhance) the gas exchange (Colberg et al., 2004) and therefore acts as a weaker (stronger) $CO_2$ sink."*

*The description of the SAM correlations are correct. A negative SAM indicates a northward displacement of the westerly winds, and higher gas exchange in the region, which moves the $CO_2$ flux towards a $CO_2$ sink (i.e stronger $CO_2$ drawdown). This is clarified in the text, which reads "Our results showed positive correlations between the $CO_2$ flux and the SAM between 30° S and 45° S (Fig. 4f) indicating stronger (weaker) $CO_2$ drawdown into the oceans during negative (positive) SAM phases."*

- Line 351: It is very likely that this signal of increased ΔpCO2 is not from the Amazon River plume, but from the waters of the North Brazil Current. A pCO2sw increase of 1.20 µatm year−1 was reported in this region by Araujo et al. (2018). The explanation for this increase is not clear, though.

***Response:*** *Thank you for the suggestion. Araujo et al. (2019) showed a $pCO_{2\,(sw)}$ increase of 1.20 $\mu atm\ yr^{-1}$, but $pCO_{2\,(atm)}$ increased at a higher rate of 1.70 $\mu atm\ yr^{-1}$ (i.e a decreasing trend in $\Delta pCO_2$). In this study we show non-significant decreasing trends in $\Delta pCO_2$ for the region evaluated by Araujo et al. (2019). These results may not explain the positive trends observed in the Amazon Plume, but add further to the discussion in the Amazon Plume region and are more representative than CARIACO (Bates et al., 2014).*

Registered Office:
Prospect Place
The Hoe, Plymouth
PL1 3DH, UK

T +44 (0)1752 633100
E forinfo@pml.ac.uk
W www.pml.ac.uk
@PlymouthMarine

Patron: James Cameron
Registered charity number 1091222.
PML is a company limited by guarantee,
registered in England & Wales,
company number 4178503

*So we have now incorporated your suggested reference of Araujo et al. (2019) into this paragraph, which reads "The Western Tropical Atlantic, in the vicinity of the Amazon Plume, also showed positive trends in $\Delta pCO_2$ and $CO_2$ flux. Previous studies have not investigated the trends in $\Delta pCO_2$ or $CO_2$ flux in the Amazon Plume, however the carbon retention in a colored ocean site (CARIACO), situated to the northwest, displayed positive trends in $pCO_{2\,(sw)}$ of $2.95 \pm 0.43\ \mu atm\ yr^{-1}$ (Bates et al., 2014). Araujo et al. (2019) identified a positive trend in $pCO_{2\,(sw)}$ of $1.20\ \mu atm\ yr^{-1}$, but a trend in $pCO_{2\,(atm)}$ of $1.70\ \mu atm\ yr^{-1}$ (i.e. decreasing $\Delta pCO_2$) for the northeast Brazilian coast, Although, the air-sea $CO_2$ flux and $\Delta pCO_2$ within the Amazon Plume region is spatially and temporally variable (Valerio et al., 2021; Ibánhez et al., 2016; Bruto et al., 2017)."*

- Line 354: Bates et al. (2014) is not in the reference list.

***Response:*** *We have checked the reference list and Bates et al. (2014) is present.*

- Line 357: Despite identifying a negative trend in both ΔpCO2 and CO2 flux, you do not explain it. For example, there are likely no significant trends in pCO2sw and the trend in ocean CO2 uptake is in response to increasing atmospheric pCO2.

***Response:*** *We have clarified the trends in the South Atlantic gyre. The uncertainties in the trends presented in Fig. B2 show these are at the limits of the uncertainties, and only just significant. We have clarified this in the text, before discussing the three studies that found significant trends (Landschützer et al., 2016; Fay and McKinley, 2013; Gregor et al., 2019) and Kitidis et al. (2017) that found no significant trend. This text now reads "The South Atlantic gyre exhibited negative trends in $\Delta pCO_2$ and the $CO_2$ flux indicating an increasing drawdown of atmospheric $CO_2$ into the ocean, which were consistent with Landschützer et al. (2016) over the period from 1982 and 2011 though the trends were at the limits of the uncertainties (Fig. B2)."*

References mentioned:

Araujo, M., Noriega, C., Medeiros, C., Lefèvre, N., Ibánhez, J. S. P., Montes, M. F., et al. (2018). On the variability in the CO2 system and water productivity in the western tropical Atlantic off North and Northeast Brazil. Journal of Marine Systems, 1, 1. https://doi.org/10.1016/j.jmarsys.2018.09.008.

Henley, S. F., Cavan, E. L., Fawcett, S. E., Kerr, R., Monteiro, T., Sherrell, R. M., et al. (2020). Changing biogeochemistry of the Southern Ocean and its ecosystem implications. Front. Mar. Sci. 7:581. https://doi.org/10.3389/fmars.2020.00581.

Ho, D. T. et al. Measurements of air-sea gas exchange at high wind speeds in the Southern Ocean: Implications for global parameterizations. Geophys. Res. Lett., 33 (2006), pp. 1-6. https://doi.org/10.1029/2006GL026817.

Liutti, C. C. et al. (2021). Sea surface CO2 fugacity in the southwestern South Atlantic Ocean: An evaluation based on satellite-derived images. Marine Chemistry, 104020. https://doi.org/10.1016/j.marchem.2021.104020.

[Figure]

Registered Office:
Prospect Place
The Hoe, Plymouth
PL1 3DH, UK

T +44 (0)1752 633100
E forinfo@pml.ac.uk
W www.pml.ac.uk
@PlymouthMarine

Patron: James Cameron
Registered charity number 1091222.
PML is a company limited by guarantee,
registered in England & Wales,
company number 4178503

Wanninkhof, R. Relationship between wind speed and gas exchange over the ocean revisited. Limnol. Oceanogr. Meth., 12 (2014), pp. 351-362. https://doi.org/10.1029/92JC00188.

**Response to Anonymous Reviewer #2**

The paper investigates variability of DpCO2 and air-sea CO2 fluxes during 2002-2018.

The paper is a follow up of two paper/published data set (Ford et al., 2021, 2022) which describe how the DpCO2 is estimated by a neural network, based on the available SOCAT cruises in the Atlantic, as a function of SST, NCP, and NPP (both derived statistically from SST and chlorophyll data). The paper examines variability in this product, as well as in derived air-sea CO2 fluxes, with a separation of the variability in seasonal, interannual and trend (so called X-11 analysis). It also attempts to statistically relate the observed (non-seasonal) variability to climate forcing of SST and biological drawdown, such as by ENSO or NAO.

*Response: Thank you for your appraisal of our manuscript and the comments that you have provided below which have improved the manuscript. Detailed responses to each comment are given below.*

Overall comments:

In some ways it is a little bit surprising that the correlations of DpCO2 and derived air-sea CO2 fluxes be discussed with SST, NCP and NPP, as these are key ingredients in which the fields of DpCO2 are constructed. This is indeed acknowledged on lines 137-145, but maybe a little more should be said on how this could limit the scope of the analysis.
On the other hand, it is interesting a posteriori to investigate the respective weight of each contribution (noting that there is probably cross-correlation between the different variables used (SST, NCP, NPP) for diagnosing DpCO2). Interestingly on inter-annual time scales, the correlations depicted on figure 3 (for DpCO2) are fairly low. They become much larger when considering the fluxes. This is, however, not surprising as there are the known relationships between the winds and the SST, NCP, NPP patterns. At the end, I was somewhat wondering what is the new information that is been brought by the study, also taking into account the short duration of the record, and thus the small number of realizations of the variability that it encompasses (I believe that this should be more clearly pointed out). At least I was not necessarily expecting the patterns of correlations with MEI (for NAO I am a little bit less convinced with a 'significant' correlation pattern only on the far southern part of the domain; and for SAM the correlation pattern seems where it is expected).

*Response: Thank you for the comments. The limitations of applying the driver's analysis to the SST, NCP and NPP fields which are a key component of estimating the $pCO_{2\,(sw)}$ are highlighted within the methods section 2.6. This limitation is discussed in the context of the results of Ford et al. (2022) that showed the SA-FNN accurately represented seasonal $pCO_{2\,(sw)}$ in the South Atlantic Ocean. By performing the analysis on $\Delta pCO_2$ estimated from in situ $pCO_{2\,(sw)}$ we provide evidence that the SA-FNN is accurately retrieving the drivers of $pCO_{2\,(sw)}$ that are present in the in situ observations. Within the text we have addressed the likely cross-correlation between data and provide reassurance that these are not major limitations through different lines of evidence, and therefore we cannot see any additional information that could be included. The approach now also includes SSS as a driving parameter, which is not used in the estimation of $pCO_{2\,(sw)}$ within the SA-FNN.*

Registered Office:
Prospect Place
The Hoe, Plymouth
PL1 3DH, UK

T +44 (0)1752 633100
E forinfo@pml.ac.uk
W www.pml.ac.uk
@PlymouthMarine

Patron: James Cameron
Registered charity number 1091222.
PML is a company limited by guarantee,
registered in England & Wales,
company number 4178503

*As given in the abstract and conclusions, the major findings of this work are that:*
*(1) Variability in biological acitivity is important for interannual and multi-year variability in the air-sea $CO_2$ flux then previously thought, and cannot be assumed to be in steady state.*
*(2) When assessing the varaibility in the oceanic $CO_2$ sink, the air-sea $CO_2$ flux should be used instead of $\Delta pCO_2$, especially on interannual and multi-year timescales.*
*(3) We provide a synoptic understanding of the drivers of the air-sea $CO_2$ flux at both seasonal and interannual timescales in the South Atlantic Ocean.*
*These results have clear implications for our current understanding of the global ocean $CO_2$ sink, the drivers of varaibility at different timescales and the contribution of biological acitivity when estimating the variability in the $CO_2$ sink on interannual and multi-year timescales.*

Not been very familiar with the X-11 approach, I was also wondering what is the frequency content of the interannual variability. In particular, I assume that with the local ('in time') definition of 'seasonal', this filters out most of the seasonal variability. On the other hand, NAO for example is quite strongly seasonally modulated. I would not expect the pattern of correlation betweenDpCO2 or derived air-sea CO2 fluxes on interannual time scales to be the same for different seasons. Could this be tested?

*Response: The frequency of the interannual variability is discussed in Pezzuli et al. (2005; see their Fig. 1) and shows the interannual component of the X-11 has frequencies all below 1 (where 1 indicates the annual cycle). Pezzuli et al. (2005) showed a 5 month running mean anomaly allowed significant amount of variability from frequencies greater than 1 (i.e annual and sub annual variability). This identified the interannual component was a more robust representation of interannual variability. The main result of Pezzuli et al. (2005) is described in the text, which reads "The analysis has been shown to be effective in the decomposition of environmental time-series (Pezzulli et al., 2005; Vantrepotte & Mélin, 2011; Henson et al., 2018), and the ability of the seasonal cycle to vary on a yearly basis, produces an interannual component that results in a robust representation of the longer-term changes in the timeseries."*

*The X-11 analysis removes the seasonal cycle through decomposition, so the effect of seaosnal modulation is actually removed. However, in addressing reviewer comments we have removed the NAO from the analysis.*

Furthermore, the issue of separation of interannual variability and trends is not that obvious with a 16-year long record. It could be particularly hard if the fields (and/or indices) present a continuous spectrum with relatively large decadal/interdecadal variability. Is it the case, and if so, how does it affect the presentation of the trends versus interannual variability. Does the presentation of trends really add much to the paper?

*Response: The time series examined here is constrained to 16 years by the availability of consistent and high-quality satellite observations from which we derive our data. Nevertheless, the period in question (2002-2018) encompasses three 'weak', one 'moderate' and one 'very strong' El Nino as well as two 'weak', three 'moderate' and one 'very strong' La Nina phases (https://psl.noaa.gov/enso/mei/). Higher frequency oscillations with multiple negative and positive phases are observed in the SAM over our time-series (Wachter et al., 2020). We feel that our analysis*

Registered Office:
Prospect Place
The Hoe, Plymouth
PL1 3DH, UK

T +44 (0)1752 633100
E forinfo@pml.ac.uk
W www.pml.ac.uk
@PlymouthMarine

Patron: James Cameron
Registered charity number 1091222.
PML is a company limited by guarantee,
registered in England & Wales,
company number 4178503

Research excellence supporting a sustainable ocean

*is therefore informative in its examination of interannual variability despite the length of the time series.*

I was also puzzled that the comment that the trend in CO2 flux were generally of lower magnitude. How does one compare the two, not being in the same unit. Is it compared to respective magnitude of interannual or seasonal variability?

***Response:*** *Thank you for highlighting this error. It is correct we cannot compare the magnitudes of $\Delta pCO_2$ and $CO_2$ flux trends which have different units and have therefore removed this comment. The sentence now reads "As with the seasonal and inter-annual analysis, the $CO_2$ flux-based trend analysis showed a greater spatial area of significant trends, when compared to $\Delta pCO_2$ (Fig. 5)."*

Minor comments

When first mentioning the X-11 analysis, it would be useful for the average reader to summarize in a few words what this method does, not just citing references.

***Response:*** *We have now added a summary of the X-11 analysis and the calculations required to decompose the timeseries into seasonal and interannual components. This text reads: "The X-11 analytical econometric tool (Shiskin et al., 1967) was used to decompose the timeseries into seasonal, interannual and residual components following the methodology of Pezzulli et al. (2005). In brief, the X-11 method comprises a three step filtering algorithm; (1) The interannual component ($T_t$) is initially estimated using an annual centred running mean, which is subtracted from the initial timeseries ($X_t$) to estimate the seasonal component ($S_t$). (2) $T_t$ is revised by applying an annual centred running mean to the $X_t$ minus $S_t$. The revised $T_t$ is removed from $X_t$ and the final $S_t$ calculated. (3) The final $T_t$ is calculated by applying an annual centred running mean to $X_t$ minus the revised $S_t$. The analysis has been shown to be effective in the decomposition of environmental time-series (Pezzulli et al., 2005; Vantrepotte & Mélin, 2011; Henson et al., 2018), that allows the seasonal cycle to vary on a yearly basis and, produces an interannual component that results in a robust representation of the longer-term changes in the timeseries."*

Line 49 : at the base and top of the boundary layer to describe the boundary layers in both media is a bit vague. Maybe add that it is the marine boundary layer that is considered.

***Response:*** *We have modified the text to clarify that the mass boundary layer discussed is at the ocean's surface. This now reads "The air-sea $CO_2$ flux is more precisely a function of the difference in $CO_2$ concentrations across the mass boundary layer at the ocean's surface, with any turbulent exchange characterised by the gas transfer velocity"*

On lines 137-145, it is mentioned that (app. A) using Henson et al (2018)'s approach yields similar results. However, I was not quite sure of what is compared (and where?) as the two analyses do not cover the same region. I am also wondering about the SOCAT data coverage in this region, especially south of the equator and away from the eastern and western boundaries. I and not 100% convinced that on interannual time scales a similar pattern emerges. Actually, that is acknowledged in App. A, whereas comment on lines 145-147 suggests the opposite.

Registered Office:
Prospect Place
The Hoe, Plymouth
PL1 3DH, UK

T +44 (0)1752 633100
E forinfo@pml.ac.uk
W www.pml.ac.uk
@PlymouthMarine

Patron: James Cameron
Registered charity number 1091222.
PML is a company limited by guarantee,
registered in England & Wales,
company number 4178503

*Response: Thank you for highlighting this contradiction, which has now been corrected. We performed the drivers analysis described in Henson et al. (2018) using in situ $pCO_{2\ (sw)}$ observations in the South Atlantic Ocean. This analysis was compared to the SA-FNN estimated $\Delta pCO_2$ drivers (Fig. 1 and 3). The seasonal drivers between the SA-FNN and the in situ analysis were consistent but on interannual timescales differences between the two analyses occurred. This is discussed in Appendix A. We have modified the main text to clarify the analysis in Appendix A was compared to the SA-FNN $\Delta pCO_2$ drivers, and that differences occurred on interannual scales likely due to the in situ timeseries averaging. This text now reads "Secondly, conducting the analysis described by Henson et al. (2018) using in situ $pCO_{2\ (sw)}$ to estimate $\Delta pCO_2$ on a per province basis (Longhurst, 1998) for the South Atlantic Ocean, yielded similar seasonal drivers to the SA-FNN (Appendix A). The interannual drivers displayed some differences however, which may be due to the spatial and temporal averaging that is required to construct the in situ timeseries."*

In 3.2, Higher correlations for fluxes, including for NCP and NPP. This is to some extent discussed later on, but not specifically for those variables. Is it expected?

*Response: Higher magnitude correlations for the $CO_2$ fluxes could be expected due to the correlations between NCP, NPP and wind speed as described in your overall comments. This is discussed in terms of NCP and the wind speed modification by the MEI in the South Atlantic gyre, referencing the results of Ford et al. (2021b) where NCP was shown to be positively correlated with wind speed. This text reads "The South Atlantic Subtropical Anticyclone has been observed to strengthen (weaken) and move south (north) during La Niña (El Niño) events. This displacement increases (decreases) wind speeds across the subtropical South Atlantic, which will enhance (weaken) gas exchange, and elevate (depress) NCP (Ford et al., 2021b)."*
*These results reinforce that NCP is important for interannual variability in the air-sea $CO_2$ flux, as this shows a biological reinforcement of the $CO_2$ flux variability.*

l. 348, I don't fully agree. There is also anthropogenic effects in the upwelled water (which has been in contact with the atmosphere 5-10 years before for a good part of it…; but that probably implies a 15 microatm difference with actual conditions)

*Response: We agree that if the deep waters that are upwelled are also subjected to an increase in $CO_2$ concentrations that roughly follows the atmospheric $CO_2$ rise, then the theoretical trend in $\Delta pCO_2$ would be ~0. We however could not find evidence to support this, and therefore have not modified the sentence, and hope you understand our reasoning.*

l. 353: CARIACO is used as a reference site. The local conditions are rather different, and strongly dependent on local upwelling (or not) conditions. I am therefore not so sure what the relation should be with the larger-scale pattern commented.

*Response: This has been clarified with the text. The inclusion of CARIACO as a reference site was to provide context for these trends in the Western Tropical Atlantic, acknowledging that the Amazon Plume is spatially and temporally variable feature. We have been made aware of the study by Araujo et al. (2019), who reported decreasing $\Delta pCO_2$ along the northeast Brazil coast, which is more representative of the Amazon Plume region.*

Registered Office:
Prospect Place
The Hoe, Plymouth
PL1 3DH, UK

T  +44 (0)1752 633100
E  forinfo@pml.ac.uk
W  www.pml.ac.uk
🐦 @PlymouthMarine

Patron: James Cameron
Registered charity number 1091222.
PML is a company limited by guarantee,
registered in England & Wales,
company number 4178503

*This new reference (Araujo et al. 2019) has been incorporated into the discussion which reads "The Western Tropical Atlantic, in the vicinity of the Amazon Plume, also showed positive trends in $\Delta pCO_2$ and $CO_2$ flux. Previous studies have not investigated the trends in $\Delta pCO_2$ or $CO_2$ flux in the Amazon Plume, however the carbon retention in a colored ocean site (CARIACO), situated to the northwest, displayed positive trends in $pCO_{2\,(sw)}$ of $2.95 \pm 0.43$ µatm yr$^{-1}$ (Bates et al., 2014). Araujo et al. (2019) identified a positive trend in $pCO_{2\,(sw)}$ of 1.20 µatm yr$^{-1}$, but a trend in $pCO_{2\,(atm)}$ of 1.70 µatm yr$^{-1}$ (i.e. decreasing $\Delta pCO_2$) for the northeast Brazilian coast, Although, the air-sea $CO_2$ flux and $\Delta pCO_2$ within the Amazon Plume region is spatially and temporally variable (Valerio et al., 2021; Ibánhez et al., 2016; Bruto et al., 2017)."*

l. 363-364: I don't understand the exact recommendation. 'Long-term' or 'inter-annual', and what is the link with the end of the sentence ', as the importance of changes in solubility and surface turbulence… increases'?

***Response:*** *We have corrected the statement to multi-year trends to be consistent between the two sentences. The recommendation is to use the $CO_2$ flux to assess multi-year trends in the ocean $CO_2$ sink, as changes in solubility and the gas transfer (which is estimated via a wind speed proxy) become significant. We have changed "surface turbulence" to "gas transfer" to be consistent with the sentence in section 5. We have also added a reference to Keppler and Landschützer (2019), to be consistent with a similar sentence within section 4.2. These sentences now read "These contradictory trends support the conclusion that $\Delta pCO_2$ is unlikely to be representative of the $CO_2$ flux over multi-year timescales. Therefore, we recommend that the $CO_2$ flux should be used to assess multi-year variability in the oceanic $CO_2$ sink, as the importance of changes in solubility and gas transfer velocity (estimated via wind speed) increases (Keppler and Landschützer, 2019)."*

l. 395-398: I did not find it easy to fully understand what has been done to the data

***Response:*** *The reanalysis of the in situ $pCO_{2\,(sw)}$ to a consistent temperature and depth dataset described in Appendix A is a rewording of the same process in section 2.1, that is applied to the in situ observations used in training the SA-FNN. We have modified the wording in Appendix A to be consistent with section 2.1. This sentence now reads "The in situ $pCO_{2\,(sw)}$ observations were obtained from SOCATv2020 (https://www.socat.info/; Bakker et al., 2016), and were reanalysed to a temperature dataset representative for a consistent and fixed depth (Reynolds et al., 2002) which is used to represent the base of the mass boundary layer. The reanalysis method used the 'fe_reanalyse_socat.py' routine within FluxEngine (Holding et al., 2019; Shutler et al., 2016), which follows the methodology of Goddijn-Murphy et al. (2015) and as used in Woolf et al. (2019) and Watson et al (2020)."*

In the discussion of 'drivers' (chapter 4) the effects of solubility and surface turbulence are mentioned. I was not exactly sure of what meant by turbulence. Does it infer to wind-induced vertical mixing?

***Response:*** *We have clarified the sentence. Turbulence in this instance refers to the small-scale motions and shear in the ocean's mass boundary layer which controls the rate of $CO_2$ exchange across the surface, which is captured by the gas transfer parameterisation. Wind speed does not directly modify the gas transfer but induces turbulence in the mass boundary layer that controls the rate of gas transfer. We have changed "surface turbulence" to "gas transfer" to be consistent with a*

[Figure]

*similar sentence in section 5. This sentence now reads "Therefore, we recommend that the CO₂ flux should be used to assess multi-year variability in the oceanic CO₂ sink, as the importance of changes in solubility and gas transfer velocity (estimated via wind speed) increases (Keppler and Landschützer, 2019)."*

    @PlymouthMarine

Patron: James Cameron
Registered charity number 1091222.
PML is a company limited by guarantee,
registered in England & Wales,
company number 4178503

[revised manuscript text omitted]

Registered Office:
Prospect Place
The Hoe, Plymouth
PL1 3DH, UK

T +44 (0)1752 633100
E forinfo@pml.ac.uk
W www.pml.ac.uk
@PlymouthMarine

Patron: James Cameron
Registered charity number 1091222.
PML is a company limited by guarantee,
registered in England & Wales,
company number 4178503

---

## Author Response (AR1)

8th June 2022.

Biogeosciences.

Dear Koji Suzuki,

Thank you for your and the reviewers' comments on our manuscript entitled 'Identifying the biological control of the annual and multi-year variations in South Atlantic air-sea CO2 flux' by Ford, Tilstone, Shutler and Kitidis. We have addressed all of the reviewers' comments and implemented these in the new version of the manuscript. We provide detailed responses to each of the reviewers' comments below. In the responses we refer to page and line numbers in the tracked changed document. We hope that you find these changes satisfactory and acceptable.

We look forward to hearing from you Yours sincerely,

**Daniel Ford**

T +44 (0)1752 633100 E forinfo@pml.ac.uk

W www.pml.ac.uk

@PlymouthMarine

**Response to Anonymous Reviewer #1**

In this study, the authors assess the importance of sea surface temperature (SST) and biological activity (i.e., photosynthesis and respiration) on sea-air CO2 fluxes in the South Atlantic Ocean on seasonal and interannual scales. They used partial pressure of CO2 in the sea surface (pCO2sw) and in the atmosphere (pCO2atm), SST, NCP, NPP, estimated essentially from satellite images, as well as wind speed from reanalysis and index of climate variability modes (i.e., ENSO, SAM, NAO). Thus, they were able to correlate the differences between pCO2sw and pCO2atm and the sea-air CO2 flux with the parameters that are involved in the seasonal and interannual variation of the carbon cycle in the Atlantic Ocean. Some aspects are explored throughout the manuscript, but its main finding is that biological activity is a more important driver of interannual variation in CO2 fluxes than previously thought. This is because biological activity is generally associated with seasonal variations in CO2 flux, while large-scale physical processes are associated with long-term variations. Therefore, I consider that both the idealization of the study and its findings are relevant to the scientific community and should be encouraged for publication. However, I raised some issues that need to be improved in order to clarify some points, mainly in the discussion and in the methods used.

**Response:** Thank you for your appraisal and support of our manuscript, and the comments that you have provided which have improved the manuscript. Detailed responses to each of your comments are given below.

**Overall:**

Essentially, all parameters used in this study are estimated. SA-FNN uses 10 gridded SOCAT pCO2sw, which generates uncertainty in the pCO2sw estimates for every 10 gridded. pCO2atm is also estimated on a global scale. Wind speed from reanalysis is expected to underestimate in situ measurements over a wide area. Therefore, you have uncertainties in the estimates of: pCO2sw, pCO2atm, SST, salinity, wind speed. What is the uncertainty propagated by these uncertainties throughout the calculation and what is the impact on the calculated trends? The authors often warn that they considered uncertainties throughout the calculations, but it is not clear which uncertainties of each parameter were introduced in this analysis. Furthermore, nothing is mentioned about numerical uncertainties. For example, what is the average uncertainty (maximum-minimum) of the calculated trends and CO2 fluxes? My suggestion is to include a section on uncertainties and limitations and a table with uncertainties for all parameters (those that are available).

**Response:** This is a good point and thank you for the suggestions on the uncertainties. In the updated manuscript, we have now included a table (Page 6) of the uncertainty values used for each of the parameters which are propagated through the calculations of  $\Delta pCO_2$  and the CO2 flux; this table (Page 6) is also given below for convenience:

| Parameter             | Uncertainty        | Reference                  |
|-----------------------|--------------------|----------------------------|
| $pCO_{2 (sw)}$        | Variable (Fig. B1) | (Ford et al., 2022)        |
| SST                   | 0.441 °C           | (Ford et al., 2021a)       |
| SSS                   | 0.1 psu            | (Jean-Michel et al., 2021) |
| $pCO_{2 (atm)}$       | 1 µatm             | (Takahashi et al., 2009)   |
| Gas transfer velocity | 20 %               | (Woolf et al., 2019)       |

Table 1: Uncertainties in the input parameters used in the Monte Carlo uncertainty propagation.

T +44 (0)1752 633100

E forinfo@pml.ac.uk

W www.pml.ac.uk

@PlymouthMarine

Following your useful comments, we have now also propagated the  $pCO_{2 (atm)}$  uncertainty within the  $\Delta pCO_2$  analysis, and SST, SSS and  $pCO_{2 (atm)}$  uncertainties within the  $CO_2$  flux analysis. The methods section 2.4 has been updated to reflect this change (Page 5 Lines 137 – 139), and all figures reproduced to reflect the new uncertainties. The low uncertainty associated with the  $pCO_{2 (atm)}$  calculation has a small impact on the overall combined uncertainty.

We have added an Appendix B (Pages 26-27) to the manuscript to show the uncertainties in the SA-FNN  $pCO_{2(sw)}$  and the associated trends as two new figures. The first figure (Figure B1; Page 26) presents the mean and standard deviation of the SA-FNN  $pCO_{2(sw)}$  uncertainty, which highlights regions where the uncertainties are higher (a) and more variable (b). This figure (Figure B1) is given here:

Figure B1 – (a) Mean SA-FNN  $pCO_2$  (sw) uncertainty between July 2002 and December 2018. Longhurst provinces (Longhurst, 1998) used within the SA-FNN training described in Ford et al. (2022). The province areas acronyms are listed as follows:western tropical Atlantic is WTRA; eastern equatorial Atlantic is ETRA; South Atlantic Gyre is SATL; Brazilian current coastal is BRAZ; Benguela Current coastal upwelling is BENG; Southwest Atlantic shelves is FKLD; South Subtropical Convergence is SSTC; sub-Antarctic is SANT and Antarctic is ANTA. (b) Standard deviation of SA-FNN  $pCO_2$  (sw) uncertainty.

The second figure (Figure B2, Page 27) addresses the uncertainty in the  $\Delta pCO_2$ , and  $CO_2$  flux trends introduced by propagating the parameter uncertainties that are given in Table 1. Instead of providing the maximum and minimum uncertainties we have mapped the uncertainties on a per pixel basis, to display where the uncertainties in the trends are higher. Figure B2 is displayed below:

Registered Office: Prospect Place The Hoe, Plymouth PL1 3DH, UK T +44 (0)1752 633100 E forinfo@pml.ac.uk W www.pml.ac.uk

@PlymouthMarine

---

## Author Response (AR2)

11th July 2022.

Biogeosciences.

Dear Koji Suzuki,

Thank you for your and the reviewers' comments on our manuscript entitled 'Identifying the biological control of the annual and multi-year variations in South Atlantic air-sea CO₂ flux' by Ford, Tilstone, Shutler and Kitidis. We have addressed all of the reviewers' comments and implemented these in the new version of the manuscript. We provide detailed responses to all of the reviewers' comments below. In the responses we refer to page and line numbers in the tracked changed document. We hope that you find these changes satisfactory and acceptable.

Thank you for your time and we look forward to hearing from you.
Yours sincerely,

[Figure]

Daniel Ford

Registered Office:    T  +44 (0)1752 633100    Patron: James Cameron
Prospect Place        E  forinfo@pml.ac.uk      Registered charity number 1091222.
The Hoe, Plymouth     W  www.pml.ac.uk          PML is a company limited by guarantee,
PL1 3DH, UK           🐦 @PlymouthMarine        registered in England & Wales,
                                                company number 4178503

[Figure]

**Research excellence supporting a sustainable ocean**

**Response to Anonymous Reviewer #1**

Whereas the authors have responded to most of my comments, I still don't fully agree with some of the statements. I however feel that the paper's clarity, aims, and presentation of main results have improved to the level that it can be published.

*Response: We thank the reviewer for the second review of our manuscript. We have addressed your remaining comments below and hope you find the revisions we have made satisfactory.*

I appreciate the drop of NAO in the discussion. Maybe the comments and the discussion of the role of the Southern annular mode (as correctly stated, partially influenced by equatorial Pacific modes of variability) in flux variability could be improved.

*Response: We understand the reviewer's point. Unfortunately, further discussion of the SAM within the manuscript is limited by our analysis not resolving correlations south of 45° S. Keppler and Landschützer (2019) investigated the control of the SAM on $CO_2$ flux variability south of 35° S and showed this effect to be more pronounced below 45° S, and therefore outside the domain of this study. We have therefore added a sentence to highlight this limitation, on Page 14 Lines 337 – 339, which reads: "It should be noted that the effect of the SAM may be more pronounced outside the domain of the present study (i.e south of 45 °S; Keppler and Landschützer, 2019).". As the significant correlations between the SAM and $CO_2$ flux in this study occupied a region between 30 °S and 45 °S, we do not feel that further discussion on the SAM would be helpful.*

My main remaining comment is with respect to the equatorial upwelling: I understand the reasoning of the response, but I still don't think that this is applicable for the equatorial Atlantic (l. 420). There is no reason for the upwelled subsurface water to have maintained constant pCO2 (age of upwelled water from the time they were at the surface somewhere in the southern Atlantic Ocean has been known since the 1980s, for example, from CFCs and 3H/3He derived ages (Reverdin et al., 1993, JGR, https://doi.org/10.1029/93JC00976). There is no doubt that the bulk of this water has been at the surface 10-15 years before being upwelled, somewhere in the southern Atlantic Ocean. Furthermore, since then, regularly acquired DIC data at some regularly-sampled equatorial stations of the FRENCH PIRATA cruises in the eastern equatorial Atlantic indicate (together with earlier cruises) an increase in time of DIC over the last two (four) decades.

*Response: We agree with the reviewer's comment, and have now modified the sentence to say that the $\Delta pCO_2$ trend in the Equatorial Atlantic should be ~0 under the assumption that the upwelled water's $CO_2$ concentrations are also increasing, due to recent contact with the atmosphere. These sentences, on Page 15 Lines 377-380, now read: "For the Equatorial upwelling, an increase in $\Delta pCO_2$ (as shown here and in Landschützer et al., 2016) is counter intuitive because there is evidence that upwelled water has recently been in contact with the atmosphere (~15 years; Reverdin et al., 1993). Dissolved inorganic carbon in these upwelled waters has been shown to increase at a similar rate to the surface waters (e.g Woosley et al., 2016). Therefore, the trend in $\Delta pCO_2$ should be ~0 with increasing $pCO_{2\ (atm)}$.".*

Registered Office:
Prospect Place
The Hoe, Plymouth
PL1 3DH, UK

T  +44 (0)1752 633100
E  forinfo@pml.ac.uk
W  www.pml.ac.uk
🐦 @PlymouthMarine

Patron: James Cameron
Registered charity number 1091222.
PML is a company limited by guarantee,
registered in England & Wales,
company number 4178503

**Response to Anonymous Reviewer #2**

The authors followed most of my and the other reviewer's suggestions, and those that were rejected were properly substantiated, in my opinion. My main concern was about demonstrating the uncertainties in the parameters used in the CO2 flux calculations and the authors inserted them in the manuscript. Another concern, shared by the other reviewer, was about a more detailed explanation of the method for assessing seasonal and interannual drivers of dpCO2 and CO2 flux, and this was duly improved as well. I just suggest a last general review of the manuscript to correct some typos, such as lines 84 (where a ";" should be ":") and 459 (where "it's" should be "it is"), if this really is grammatical errors, as I am not very knowledgeable about English grammar.

***Response:*** *We thank the reviewer for the second review of our manuscript, and are glad that are revisions to the manuscript were satisfactory. We have now corrected these two typographical errors and we have checked the complete manuscript for typographical and grammatical errors.*

**References**

Keppler, L. and Landschützer, P.: Regional Wind Variability Modulates the Southern Ocean Carbon Sink, Sci. Rep., 9, 1–10, https://doi.org/10.1038/s41598-019-43826-y, 2019.

Reverdin, G., Weiss, R. F., and Jenkins, W. J.: Ventilation of the Atlantic Ocean equatorial thermocline, J. Geophys. Res., 98, 16289, https://doi.org/10.1029/93JC00976, 1993.

Woosley, R. J., Millero, F. J., and Wanninkhof, R.: Rapid anthropogenic changes in $CO_2$ and pH in the Atlantic Ocean: 2003-2014, Global Biogeochem. Cycles, 30, 70–90, https://doi.org/10.1002/2015GB005248, 2016.

Registered Office:
Prospect Place
The Hoe, Plymouth
PL1 3DH, UK

T  +44 (0)1752 633100
E  forinfo@pml.ac.uk
W  www.pml.ac.uk
@PlymouthMarine

Patron: James Cameron
Registered charity number 1091222.
PML is a company limited by guarantee,
registered in England & Wales,
company number 4178503

---

## Author Response (AR3)

Research excellence supporting a sustainable ocean

15th August 2022.

Biogeosciences.

Dear Koji Suzuki,

Thank you for accepting our manuscript entitled 'Identifying the biological control of the annual and multi-year variations in South Atlantic air-sea $CO_2$ flux' by Ford, Tilstone, Shutler and Kitidis for publication in Biogeosciences. We have addressed your technical corrections and implemented these in the next version of the manuscript. We hope you find these changes outlined below satisfactory.

We look forward to hearing from you
Yours sincerely,

[Figure]

Daniel Ford

Registered Office:
Prospect Place
The Hoe, Plymouth
PL1 3DH, UK

T  +44 (0)1752 633100
E  forinfo@pml.ac.uk
W  www.pml.ac.uk
   @PlymouthMarine

Patron: James Cameron
Registered charity number 1091222.
PML is a company limited by guarantee,
registered in England & Wales,
company number 4178503

[Figure]

**Response to Koji Suzuki**

Dear Dr. Ford,

It is a great pleasure to accept your manuscript for publication in Biogeosciences. Thank you very much for responding to the reviewers' comments so carefully. Before it is published in the journal, please consider my minor technical comment:

Regarding the format of Reference in Table 1, the following may be more common – for example, Ford et al. (2022), not (Ford et al., 2022).

Thank you again for choosing Biogeosciences as an outlet for your excellent work.

Kind regards,

Koji Suzuki
Associate Editor

***Response:*** *Thank you for accepting our manuscript for publication in Biogeosciences. We have now corrected the formatting of the references in Table 1 as suggested.*

Registered Office:
Prospect Place
The Hoe, Plymouth
PL1 3DH, UK

T  +44 (0)1752 633100
E  forinfo@pml.ac.uk
W  www.pml.ac.uk
   @PlymouthMarine

Patron: James Cameron
Registered charity number 1091222.
PML is a company limited by guarantee,
registered in England & Wales,
company number 4178503